# Global-regional nested simulation of particle number concentration by combing microphysical processes with an evolving organic aerosol module

Xueshun Chen[1,3], Fangqun Yu[6], Wenyi Yang[1,3], Yele Sun[1,2,3], Huansheng Chen[1], Wei Du[1,7], Jian Zhao[1], Ying Wei[1,4], Lianfang Wei[1,3], Huiyun Du[1], Zhe Wang[1], Qizhong Wu[5], Jie Li[1,3], Junling An[1,2], Zifa Wang[1,2,3]

[1]The State Key Laboratory of Atmospheric Boundary Layer Physics and Atmospheric Chemistry, Institute of Atmospheric Physics, Chinese Academy of Sciences, Beijing 100029, China
[2]College of Earth and Planetary Sciences, University of Chinese Academy of Sciences, Beijing 100049, China
[3]Center for Excellence in Regional Atmospheric Environment, Institute of Urban Environment, Chinese Academy of Sciences, Xiamen 361021, China
[4]Institute of Urban Meteorology, China Meteorology Administration, Beijing 100089, China
[5]College of Global Change and Earth System Science, Beijing Normal University, Beijing 100875, China
[6]Atmospheric Science Research Center, State University of New York at Albany, New York 12203, USA
[7]Institute for Atmospheric and Earth System Research/Physics, Faculty of Science, University of Helsinki, Helsinki 00014, Finland

*Correspondence to*: Zifa Wang (zifawang@mail.iap.ac.cn)

**Abstract**

Aerosol microphysical processes are essential for the next generation of global and regional climate and air quality models to determine the particle size distribution. The contribution of organic aerosol (OA) to particle formation, mass and number concentration is one of the major uncertainties in current models. A new global-regional nested aerosol model was developed to simulate detailed microphysical processes. The model combined an advanced particle microphysics (APM) module and a volatility basis-set (VBS) organic aerosol module to calculate

the kinetic condensation of low volatile organic compounds and equilibrium partitioning of semi-volatile organic compounds in a 3-dimensional (3-D) framework using global-regional nested domain. In addition to the condensation of sulfuric acid, equilibrium partitioning of nitrate and ammonium, and the coagulation process of particles, the microphysical processes of the organic aerosols are realistically represented in our new model. The model uses high-resolution size-bins to calculate the size distribution of new particles formed through nucleation and subsequent growth. The multi-scale nesting allows the model to use high resolution to simulate the particle formation processes in the urban atmosphere in the background of regional and global environments. Using the nested domains, the model reasonably reproduced the OA components from analysis of Aerosol Mass Spectrometry (AMS) measurements by Positive Matrix Factorization (PMF) and the particle number size distribution (PNSD) in Megacity Beijing during a period of about a month. Anthropogenic organic species accounted for 67% of the OA of secondary particles formed by nucleation and subsequent growth, significantly larger than that of biogenic OA. Over the global scale, the model well predicted the particle number concentration in various environments. The microphysical module combined with VBS simulated the universal distribution of organic components among the different aerosol populations. Model results strongly suggest the importance of anthropogenic organic species in aerosol particle formation and growth at polluted urban sites and over the whole globe under the influence of anthropogenic source areas.

**Key words:** IAP-AACM+APM, VBS, organic aerosol, particle number concentration

## 1Introduction

The increased concentrations of atmospheric aerosol particles caused by anthropogenic activities have become an important scientific issue due to their significant climate forcing and health effects (Twomey, 1977; Albrecht, 1989; Charlson et al., 1992; Donaldson et al., 2002; Tsigaridis et al., 2006; IPCC, 2013) in global and regional scales. These effects depend on aerosol size, composition, and mixing state. The direct influence of aerosols on climate is their scattering of solar radiations largely determined by the key properties of aerosols mentioned above

(IPCC, 2013). The indirect effects of aerosols are driven by their ability in affecting cloud microphysical properties and precipitation processes through serving as cloud condensation nuclei (CCN), which is highly dependent on CCN number concentrations (Dusek et al., 2006). Ultrafine particles, though having lower mass concentration, have larger health effect due to their easier penetration and higher number concentrations (Delfino et al., 2005; Kumar et al., 2014). Therefore, it is crucial to gain deep insight into the life cycle of aerosol particles and quantify their sources not only in mass concentration but also in their number concentration.

There are two sources of atmospheric aerosols: direct emissions from primary sources and secondary formation processes (Seinfeld and Pandis, 2006). Mineral dust particles over desert regions and sea salt particles over oceans are the two major natural sources contributing to the particle mass and number concentration regionally (Textor et al., 2006). Anthropogenic activities, such as fossil fuel combustion and biomass burning, can directly emit particles and they are the most significant contributors to the aerosols since the industrial revolution (IPCC, 2013). The physical and chemical properties of these primarily emitted particles can be modified by condensation, coagulation, and chemical reactions in the atmosphere (Seinfeld and Pandis, 2006). In addition, new particle formation (NPF) is found to be an important contributor to aerosol particles in global various environments (Holmes, 2007; Yu et al., 2008; Yu and Luo, 2009; Kulmala et al., 2013). Field observation studies also have demonstrated that the new particle formation can significantly increase CCN number concentrations (Kuang et al., 2009; Wiedensohler et al., 2009; Yue et al., 2011). It is necessary to reasonably represent primary emission，their microphysical aging, and new particle formation process in 3-D models.

During the past two decades, there has been numerous models incorporating microphysical module to describe the particle formation processes (e.g., Binkowski and Shankar, 1995; Jacobson, 1997; Stier et al., 2005; Bergman et al., 2012). However, large uncertainties still exist due to the complication of processes and the mechanisms not well understood. Intercomparison and evaluation of global aerosol models indicate that constraint of size-resolved primary emission and improved understanding of

secondary formations are required to improve the ability of model in simulating particle number size distribution (Mann et al., 2014). Spracklen et al. (2005) find that the assumption of the size distribution has a large impact on particle number concentrations in the boundary layer. The comparison between the simulation and the Single Particle Soot Photometer (SP2) measurements suggests that the model has large bias in simulating the number size distribution of black carbon particles (Reddington et al., 2013). Significant improvements in the simulation of the particle number concentration and aerosol optical properties were achieved by using an optimized size distribution of primary particles in polluted atmosphere over areas with large emissions (Zhou et al., 2012, 2018). Much work remains to reduce the uncertainty associated with primary emissions, especially over primary particles dominated regions in terms of particle number concentration, like China.

The main source of uncertainty in simulating new particle formation at regional and global scales can be attributed to the nucleation mechanism and particle growth rates unexplained. Although sulfuric acid has been identified as a major component and plays a central role in nucleation (Yu and Turco, 2001; Boy et al., 2005; Kirkby et al., 2011), alone it could not explain the new particle formation rates (Wang et al., 2013; Kulmala et al., 2013). Recent studies revealed that certain organic vapors are involved in the particle nucleation (Metzger et al., 2010; Zhang et al., 2012; Yao et al., 2018) and contribute much to the particle growth (Kulmala and Kerminen, 2008; Tröstl et al., 2016). It is no doubt that reasonable representation of organic aerosol (OA) is crucial for aerosol models to realistically simulate new particle formation and growth. However, it is still an open question which organic species are possibly involved in new particle formation process. Even the chemical composition and the sources of OA are still uncertain as they contain large number of compounds (Goldstein and Galbally, 2007). Up to now, OA is still the least understood one among the components of aerosols (Kanakidou et al., 2005; Hallquist et al., 2009). Clearly, the OA representation is the major uncertainty contributing to the huge gap in elucidate particle formation processes.

In recent years, much progress has been achieved in simulating the formation of

OA and secondary organic aerosol (SOA). The two product (2P) model recommended by Odum et al. (1996) had been widely used in 3-D models to describe SOA formation process empirically. The volatility basis-set (VBS) approach was recently developed to represent the oxidation of primary OA (POA) and SOA and the partitioning of OA in different volatilities between gas phase and aerosol phase (Donahue et al., 2006). Many regional models have used VBS to simulated OA and SOA (Shrivastava et al., 2008; Fountoukis et al., 2011; Ahmadov et al., 2012; Zhao et al., 2016; Han et al., 2016). However, application of VBS in global models is limited for the large number of tracers required and the uncertainty of the involved parameters (Farina et al., 2010; Hodzic et al., 2016). There are even fewer applications of this unified framework in 3-D global aerosol models to calculate the processes of particle formation. Among the second phase AeroCom aerosol microphysical models, simplified parameterization and two-product method are the mostly used schemes to represent SOA (Mann et al., 2014). Recently，there has been some models with VBS incorporated in their microphysical module to simulate aerosol microphysical formation process. Patoulias et al. (2015) developed a new aerosol dynamics model with VBS and explored the contribution of SOA with different volatility to particle growth in different stages, but the 3-D modeling was not presented. By assuming equilibrium partitioning for all volatility bins, Gao et al. (2017) implemented VBS in an aerosol microphysics model and examined the effect of semi-volatile SOA on the composition, growth, and mixing state of particles. Their simulation of box model suggested that the volatility of organic compounds simulate rather different mixing states from those simulated by coagulation process alone in the scheme treating primary emission of organics as nonvolatile. Matsui (2017) represented aerosol size distribution with a two-dimensional sectional method in a global aerosol model coupled with the VBS scheme, but the size-bin resolution is not high enough to well resolve the growth of new particles.

To our knowledge, there is currently scarce 3-D modeling study using VBS to account for both (1) the kinetic condensation of low-volatile organics and re-evaporation of semi-volatile organics and (2) the size-resolved kinetics of the mass

transfer for new particles. In addition, the particle formation in the polluted atmosphere was not well understood (Kulmala et al., 2016; Wang et al., 2017; Chu et al., 2019). Over the urban areas in northern China, observation and modeling studies indicate that anthropogenic SOA contributes a larger fraction to OA than that of biogenic one and play an significant role in particle formation (Yang et al., 2016; Guo et al., 2020; Han et al., 2016; Lin et al., 2016). Simultaneously calculating both anthropogenic and biogenic SOA in microphysical models with high resolution is crucial to resolve the particle formation processes over the urban areas. Furthermore, the previous studies focusing on the sensitivity of particle number concentration to primary emission were based on models without considering the detailed microphysics of organic species (e.g., Spracklen et al., 2006; Chang et al., 2009; Chen et al., 2018; Zhou et al., 2018). Therefore, it is urgently needed to establish a 3-D modeling framework of VBS with an aerosol microphysics module with high size-bin resolution to simulate the particle number size distribution and explore the uncertainties associated with the treatment of primary emission.

In our previous work, a regional model with detailed microphysical processes has been developed to improve the new particle formation in summer in Beijing (Chen et al., 2019). In this study, we extend our work to the global scale and doing so to establish a new aerosol model by coupling a VBS organic aerosol scheme with a particle microphysics module in a global-regional nested model. The model performance was evaluated against the measurements at a tower and the collected dataset from published papers. In addition, the model's sensitivity to the size distribution of primary emission and volatility distribution of POA are explored to understand and quantify the uncertainties associated. The new modeling framework can provide a useful tool to simulate aerosol microphysical process in both global and regional scales. The description of model and its development method are introduced in Sect.2. The experiments setup and model input are detailed in Sect.3. The observed data used for evaluating model performance are described in Sect.4. The model results and simulation analysis are presented in Sect.5. The conclusions and discussions are summarized in Sect.6.

## 2 Model description

### 2.1 Host model

The host model employed in this study is the Atmospheric Aerosol and Chemistry Model developed by Institute of Atmospheric Physics, Chinese Academy of Sciences (IAP-AACM). The IAP-AACM is a 3-D atmospheric chemical transport model treating chemical and physical processes for gases and aerosols in global and regional scales using multi-scale domain-nesting technique (Wang et al., 2001; Li et al., 2012; Chen et al., 2015). The model has been successfully used to explore mercury transport (Chen et al., 2015) and simulate the global and regional distribution of gaseous pollutants, aerosol components (Du et al., 2019; Wei et al., 2019). The calculation of some modules in the model has also been optimized recently (Wang et al., 2017, 2019). The model calculates 3-D advection (Walcek, 1998), turbulent diffusion (Byun and Dennis, 1995), gas phase chemical reactions (Zaveri and Peters, 1999), dry deposition at the surface (Zhang et al., 2003), aqueous reactions in the cloud and wet scavenging (Stockwell et al., 1990), and heterogeneous chemical process (Li et al., 2012). The partition of nitric acid and ammonia into particle phase to form nitrate and ammonium are simulated using a thermodynamic equilibrium model (Nenes et al., 1998). The model calculates the online emission of dimethyl sulfide (DMS) (Lana et al., 2011), sea salt (Athanasopoulou et al., 2008) and dust (Wang et al., 2000; Luo and Wang, 2006). The simulation results of IAP-AACM have been evaluated against a comprehensive observation dataset and compared with other model results. The model showed good performance in reproducing global aerosol components (Wei et al., 2019).

### 2.2 APM module

Advanced Particle Microphysics (APM) module is an aerosol module using the sectional method to represent particle number size distribution. APM has been coupled to several 3-D models, such as GEOS-Chem (Yu and Luo, 2009), WRF-Chem (Luo and Yu, 2011), and NAQPMS (Chen et al., 2014). In APM, there are two types of aerosol particles: one is secondary particles (SPs) and the other is primary particles (PPs) with a secondary species coating. The definitions of SPs and

PPs in our model are different from secondary aerosol and primary aerosol commonly used in the community. SPs indicate their source is the nucleation and the subsequent growth of newly nucleated particles whereas PPs are from the direct emission. PPs include dust particles, sea salt particles, organic carbon (OC) particles, and black carbon (BC) particles. The APM has a high size-bin resolution to accurately describe the formation and growth processes of SPs (composed of sulfate, nitrate, ammonium, and organic compounds). SPs are represented by 40 size bins from 0.0012 μm to 12 μm in dry diameter. Among the PPs, representation of BC and OC are updated from a modal method in the original version (Yu and Luo, 2009) to a size-bin scheme in the revised version (Chen, 2015). Dust particles in 0.03–50 μm are represented by 4 size bins and sea salt particles in 0.0012–12 μm are represented by 20 size bins. SPs are assumed to be internally mixed and PPs are assumed to be consisted of a primary core and coating species. SPs and PPs of different categories are externally mixed with each other. In addition to the primary core, the coated species are explicitly simulated in APM.

The basic microphysical processes in APM include nucleation, condensation/evaporation, coagulation, and thermodynamic equilibrium partition. The nucleation scheme is the ion-mediated nucleation (IMN) (Yu, 2006, 2010) physically-based and constrained by laboratory data, which have predicted reasonable distributions of global nucleation (Yu et al., 2008). Due to very low saturation vapor pressure, the condensation of $H_2SO_4$ is explicitly calculated. The semi-volatile inorganic species (nitrate and ammonium) and secondary organic species are simulated through equilibrium partitioning. The bulk mass concentrations of coating species are tracked to reduce the computational cost and the corresponding tracers used are defined as BC sulfate, OC sulfate, sea salt sulfate, and dust sulfate, respectively. For coagulation, APM not only calculates the self-coagulation of sea salt particles, BC particles, OC particles, and SPs, and but also considers the coagulation scavenging of SPs by four types of PPs. Yu (2011) has further developed APM to explicitly calculate the co-condensation of sulfuric acid and low-volatility secondary organics gas (LV-SOG) on the secondary and primary particles. In the scheme, the

production rate of LV-SOG and the semi-volatile OA input to APM are simulated with the extended two-product SOA formation model. For high calculation efficiency, a pre-calculated look-up table of coagulation kernels is used in the coagulation module. The numerical scheme used is from Jacobson et al. (1994). More details on microphysical processes of APM can be found in the study of Yu and Luo (2009).

### 2.3 VBS module

To reproduce the formation and evolution of OA, a 1.5-D VBS approach (Koo et al., 2014) based on 1-D VBS framework but accounting for changes in the oxidation state and volatility of OA in the 2-D VBS space is coupled to the model. Both secondary and primary organic aerosols are distributed in five volatility bins ranging from $10^{-1}$ to $10^{3}$ μg/m$^{3}$ in saturation concentration (C*) at 298 K, and temperature dependence of C* is calculated by the Clausiuse-Clapeyron equation (Sheehan and Bowman, 2001). The compounds distributed in the lowest bin with C* less than $10^{-1}$ μg/m$^{3}$ represents the effectively nonvolatile OAs and they are considered as low-volatile organic compounds almost partitioned to the particulate phase in our model. The compounds in other four bins, i.e., C* = {$10^{0}$, $10^{1}$, $10^{2}$, $10^{3}$} μg/m$^{3}$, are defined as semi-volatile organic compounds that can be partitioned between the gas and particulate phase by equilibrium assumption (Donahue et al., 2009). To track the oxidation state of OA, four basis sets are used in the scheme: two-basis sets for chemically aged OA from anthropogenic and biogenic sources, and two-basis sets for freshly emitted OA from anthropogenic sources and biomass burning. The molecular properties for primary OA (POA) and SOA in each volatility bins are provided by the parameters calculated by 2-D volatility scheme (Donahue et al., 2011, 2012).

In this VBS module, gas phase organic compounds can be aged by extremely reactive hydroxyl radicals (OH) and other oxidants. Volatile organic precursors of SOA in this study include compounds with terminal olefin carbon bond (R−C = C), internal olefin carbon bond (R−C = C−R). The associated species in the model are terpenes, isoprene, and aromatics. Aging of POA by OH is at a reaction rate of $4\times10^{-11}$cm$^{3}$·molecule$^{-1}$·s$^{-1}$ (Robinsonet al., 2007). In the calculation, a conception of "partial conversion" is used, i.e., the oxidation products are a mixture of POA and

oxidized POA (OPOA) in the adjacent lower volatility bins (Koo et al., 2014). In addition, the multigenerational oxidation processes of intermediate VOCs (IVOCs) with OH radicals at a rate constant of $4\times10^{-11}cm^3\cdot molecule^{-1}\cdot s^{-1}$ are taken into account in SOA formation. IVOCs emission was put into the bin of $10^4 \mu g/m^3$ saturation concentration. The VBS module in this study does not consider OA formation through aqueous-phase/heterogeneous reactions although their importance is suggested in some studies (e.g., Liu et al., 2012; Ervens et al., 2014; Lin et al., 2014). SOA generated from VOCs and IVOCs and anthropogenic OPOA are assumed to be further oxidized by OH radical at an aging rate of $2\times10^{-11}$ $cm^3\cdot molecule^{-1}\cdot s^{-1}$ based on the work in Koo et al. (2014). The volatilities of multi-generation oxidation products decrease and move down to the adjacent bin with an order of magnitude lower volatility (Donahue et al., 2006). The model has 32 pairs of semi-volatile compounds including organic gases (OGs) and the corresponding OAs through equilibrium partitioning. Plus the 8 groups of low-volatility OAs, the model has 40 groups of OAs and 32 groups of OGs in total. Detailed information on this VBS module can be found in Koo et al. (2014) and Yang et al. (2019).

**2.4 Model development**

In our previous work, the VBS module has been combined with APM to improve the simulation on new particle formation process in our regional model (NAQPMS+APM, Chen et al., 2019). Here, we use the similar method to couple the VBS and APM into the global model, i.e., IAP-AACM. The newly developed model is named IAP-AACM+APM. In the model, not only the basic microphysical processes aforementioned but also the condensation of LV-SOG and equilibrium partition of semi-volatile OA (SV-OA) are calculated following the approaches described in Yu and Luo (2009) and Yu (2011). In addition to the tracers of OAs and OGs mentioned above, a new tracer for LV-SOG is tracked in IAP-AACM+APM. The sources of sulfuric acid and LV-SOG are photochemical reactions. Their production rates are calculated by CBM-Z and VBS module, respectively. The production rate of LV-SOG is equivalent to that of the lowest bin OAs in VBS module. For simplicity and computing efficiency, the condensation of LV-SOG on SPs of different sizes is

calculated along with $H_2SO_4$ and the low volatile SOA (LV-SOA) on SPs are merged into one bulk tracer (SP-LV). When necessary, SP-LV is redistributed to size-bins according to the surface area of particles. The condensation of LV-SOG on PPs (i.e., dust, sea salt, BC, and OC particles) is calculated in the same way as $H_2SO_4$. The amount of LV-SOA coated on these particles are defined as dust-LV, salt-LV, BC-LV and OC-LV. In this way, LV-SOAs are distributed approximately proportional to the aerosol surface area. The semi-volatile SOA (SV-SOA) partitioned to SPs in each bin and the coatings on PPs are assumed to be proportional to the corresponding LV-OA mass. For OC particles, the coated SV-SOA depends on both OC-LV and POC. The SV-SOA input to APM is the total mass concentration of 32 groups of semi-volatile OAs in VBS module. The partition of this part of OA is similar with that of equilibrium partition theory (Pankow, 1994a,b; Odum et al., 1996). By using the treatments above, the different microphysical behaviors of OAs with different volatilities are reasonably simulated. The dry deposition at the surface level and wet deposition by precipitation of LV-SOG are modeled using same scheme as $H_2SO_4$. The dry deposition and wet scavenging of the coated LV-OA associated with SPs and PPs are calculated using the same scheme as the sulfate coated on PPs (Yu, 2011).

The tracers associated aerosol microphysical processes in IAP-AACM+APM are listed in Table 1. The number of newly added tracers in IAP-AACM+APM is 129 compared to IAP-AACM; and therefore, the computing time of 3-D advection and turbulent diffusion is nearly doubled comparing to IAP-AACM. Among the modules in IAP-AACM+APM, gas phase reaction module and the microphysical module are most time-consuming. The newly-developed processes in IAP-AACM+APM do not add much computing time. The total computing time of IAP-AACM+APM is less than that twice of IAP-AACM and it is acceptable. The aerosol microphysical module combining VBS with APM in this study can be used in other 3-D models.

## 3 Model configuration and experiments setting

### 3.1 Model domain and model inputs

In this study, we use two nested modeling domains for one-year simulation in 2010, with the first domain covering whole globe at 1 degree resolution, the second

domain covering east Asia at 0.33 degree resolution. The model has 20 vertical layers and the top layer is at 20 km. The simulation from December 1st , 2009 to December 31st, 2010 was used for annual mean analysis and the first 1-month of simulation was spin-up time and not used in analysis. In addition, a case study in 2015 using three nested domains with the third domain of 0.11 degree resolution was conducted to evaluate the model performance in simulating OA components and particle number size distribution at a typical urban site. The model domains are shown in Fig.S1.

The meteorological parameters input to IAP-AACM+APM were simulated by the global version of Weather Research and Forecasting (WRF) model (Skamarock et al., 2008). The initial and boundary conditions of WRF was provided by Final Analysis (FNL) datasets from the National Centers for Environmental Prediction (NCEP) (https://rda.ucar.edu/datasets/ds083.2). The temperature, humidity, wind speed, and pressure in WRF were nudged to FNL datasets. Gridded emission inventory used in IAP-AACM+APM was an integrated dataset from a publicly datasets (https://edgar.jrc.ec.europa.eu/htap_v2/index.php) and the multi-resolution emission inventory for China (MEIC) (http://www.meicmodel.org).

### 3.2 Experiments setting

One base experiment and four sensitivity experiments are used in our study. The sensitivity experiments involving size distribution of primarily emitted particles, including BC and POC, and the volatility distribution of POA, are designed to investigate the impact of these factors on the particle number concentration. Table 2 lists the experiments used in this study. In the BASE experiment, the volatility distributions of POA from vehicles and biomass burning are based on the chamber studies (May et al., 2013a,b,c); the factors of other POA emissions are from the estimation of Robinson et al. (2007). In the LV_POA and HV_POA experiment, quartiles of the above mentioned distribution factors are used. In the OCD0.5 and PPD0.5 experiment, the geometric mean diameter is set half as the ones used in BASE experiment for POC, both BC and POC, respectively.

## 4 Observation data

The hourly observation of OA and particle number size distribution (PNSD) in

Beijing is used to evaluate the model performance in the typical urban environment. The observation site is located at the Tower Branch of the Institute of Atmospheric Physics (IAP), Chinese Academy of Sciences (CAS) (39°58′N, 116°22′E). The details of the observation site were described in Sun et al. (2015). The observation period was from 22 August to 30 September, 2015. Organics aerosol compositions were measured by a high-resolution aerosol mass spectrometer (HR-AMS, Aerodyne Research Inc.) and an aerosol chemical speciation monitor (ACSM, Aerodyne Research Inc.) at ground level and 260 m, respectively (Zhao et al., 2017). Using positive matrix factorization (PMF) algorithm (Paatero and Tapper, 1994; Paatero, 1997), organic aerosol (OA) were separated into hydrocarbon-like OA (HOA) and oxygenated OA (OOA). The detailed evaluation of PMF results was given in Zhao et al. (2017). PNSD from 15 to 685 nm at ground level and 260m on the 325m meteorological tower were measured by using two scanning mobility particle sizers (SMPSs). More details on the observation can be found in the published paper (Du et al., 2017). In model evaluation, the observed PNSD were mapped to the defined size bins of SPs in APM. OC concentrations in 2010 from the Interagency Monitoring of Protected Visual Environments (IMPROVE) network (http://vista.cira.colostate.edu/improve) and the China Atmosphere Watch Network (CAWNET) reported by Zhang et al. (2008) were used to compare with the simulated OC of our model. In addition, a list of surface observations of particle number concentration having at least one full year measurements was compiled to check model performance. Table S1 gives the compiled mean concentrations of condensation nuclei larger than 10 nm (CN10) and the corresponding station information from published papers.

## 5 Results

### 5.1 PNSD and aerosol components of SPs in Beijing

The simulated OA concentration is compared with the results of the PMF analysis of the AMS measurements before evaluating the simulated PNSD in Beijing. Here, HOA and OOA components by the PMF analysis have been compared with the simulation assuming they are primary and secondary components of OA, i.e., POA

and SOA, respectively. Affected by local emission sources (e.g., traffic emission and cooking emission), the observed values of OA were not representative at the ground level. Therefore, only HOA and OOA at the height of 260 m were used for comparison. The third-domain results at 0.11 degree horizontal resolution with the other configurations same with the base experiment were extracted for the analysis and comparison. First, BC simulations were compared with the observations (shown in Fig.1a), considering that BC is a passive tracer and it is generally co-emitted with POA. Since BC is only influenced by emissions, transport and deposition, the agreement between model and observations in Fig.1a suggests the model represented these processes reasonably well. Fig.1b and 1c show the comparison of the simulated and the observed hourly OA components at the 260m height. The comparison in Fig.1 highlights the good skill of model in capturing the variation of POA and SOA in our 3-D framework with VBS. Although the measurements at higher level were not susceptible to local emissions, the observed OA concentrations were inevitably influenced by the sources near the measurement site. For example, cooking emitted OA, assumed as a part of HOA here, have been identified as an important contributor to OA (Zhao et al., 2017). Moreover, the nearby traffic emissions would have large influences on the observed OA concentrations at the measurement site (Sun et al., 2015). Same as BC, the temporal variation of POA was mainly influenced by emissions, transport and deposition, the disagreement between the simulated POA and the observed HOA can largely be attributed to the emissions. In additions, the PMF analysis has its own uncertainties and deficiencies (Ulbrich et al., 2009). As a result, some observed values of HOA were not reproduced by IAP-AACM+APM. By contrast, most of the predicted SOA and their temporal variation were consistent with the observation of OOA although their concentrations were partially underestimated and some peaks were high. The correlation coefficient between the simulated SOA and observed OOA was 0.52. Overall, our model well simulated the POA and SOA concentrations.

During the past decades, many field observations have been conducted to study the characteristics of PNSD in Beijing (Wehner et al., 2004; Wu et al., 2007, 2008;

Wang et al., 2015). The NAQPMS model has been used to explain the evolution of PNSD in winter in Beijing (Chen et al., 2017). However, 3-D modeling study on these issues are still limited (Kulmala et al., 2016; Wang et al., 2016). Here, the observed PNSD at 260m height is used to evaluate the model performance. Fig.2 shows the comparison of simulated PNSD with the observations. In Fig.2, the model well reproduced the evolution of PNSD at the height of 260m at the measurement site. In the observation, there are five cycles of conversion from clean days to pollution days. Once the pollution episode was over, an obvious new particle formation event occurred, such as the events in September 3, 12, 19 and 25. When the pollution level increased, the PNSD shifted to end of large diameter. The model well captured the new particle formation events and the growth of particles in the pollution episode mentioned above. Because the atmosphere at higher level was not susceptible to local sources, the observation was more representative than that at the ground level. The number concentration of particles from 100 nm to 1000 nm was nicely reproduced, with normalized bias less than 40% and correlation coefficient being 0.70. The consistency between simulation and observation suggests the good performance of model in producing reasonable number concentration of regional aerosol particles, especially in the climate-relevant size range. However, the number concentration of particles from 15 nm to 25 nm was overestimated. On one hand, the measurements have analytical errors (Du et al., 2017). On the other hand, the model has several uncertainties. First, the model used the monthly mean emissions and therefore could not simulate the diurnal variation of traffic emission. In addition, the size distribution of primary emissions does not meet the assumed lognormal distribution. For example, traffic sources emit smaller particles than industrial sources (Paasonen et al., 2013; Kumar et al., 2014). Second, the nucleation scheme also has uncertainties (Zhang et al., 2010; Yu et al., 2018). For all this, the main features of new particle formation events and the growth of particles were captured by the model. Generally, our model produced the aerosols of real atmosphere and the simulation results were reasonable.

The reasonable performance of our model in simulating OA components and PNSD gives us the confidence to further analyze the composition of newly formed

particles through nucleation and subsequent growth, i.e. SPs in our model. Fig.3a shows the simulated mean contribution of sulfate, nitrate, ammonium, and OA to the mass concentration of SPs in September. Fig.3b shows the contribution of LV anthropogenic OA (LV-AOA), LV biogenic OA (LV-BOA), SV biogenic OA (SV-BOA), and SV anthropogenic OA (SV-AOA) to the mass concentration of OA in SPs. Fig.3a demonstrate that OA was the major component of SPs, followed by sulfate, nitrate, and ammonium. Among the components of OA in SPs, AOA accounted for 67%, significantly larger than the 33% of BOA, suggesting the dominant role of AOA in particle growth. In terms of volatility, LV-OA took up 67%, in which LV-AOA was responsible for 50% and LV-BOA for 17%. Our model calculated the gas-phase concentration of LV-SOG and the kinetic condensation of them on size-resolved SPs. The large fraction of LV-AOA in OA of SPs indicates their important role in the growth of SPs. Further, LV-AOA is an indicator of aged atmosphere and its large contribution to OA suggested the influence of regional transport of OA and precursors of OA from surrounding areas to Beijing. The aging and growth during the lifetime of SPs in the atmosphere could greatly enhance their regional impact. In addition to the local emissions of OA precursors (Guo et al., 2014), our results also highlight the importance of regional sources of OA precursors in the growth of new particles.

### 5.2 Global and regional distribution of OA

There are two important characteristics of OA that influence particle growth and particle number concentration: (1) the concentration of OA and (2) the condensation behavior of OA. The concentration of OA is dependent of the OA sources and sinks. The condensation behavior of OA is closely related with the separation of POA and SOA and their volatility distribution. Therefore, these properties of OA are given as the background to discuss the global and regional particle number concentration. Fig.4 shows the surface distribution of OC concentration and the fraction of secondary OC (SOC) in the Base experiment. In our model, OA is formed by primary emission and the partitioning of gas-phase species onto preexisting organic aerosol. Therefore, the distribution of OC is well correlated with the amount of primary emission and SOA

precursors. Globally, the high concentrations of OC are located in the continent regions with large emissions. Over China and India, OC concentration can be above 10 μg/m$^3$ due to the high emissions from intense anthropogenic activities. In the tropical region of Africa, OC concentrations are larger than 5 μg/m$^3$ due to the biomass burning. Over America and Europe, OC concentrations are below 3 μg/m$^3$. The model well reproduced this spatial difference reflected by the observations in America and China. The highest concentrations are located in central-eastern China. In the second domain (shown in Fig.4c), the highest concentrations of simulated OC can be above 15 μg/m$^3$ over some areas in Sichuan Basin and North China Plain. The observed values of OC are underestimated at most sites in China. Firstly, the difference of time between simulation and observation, especially the emission change, could lead to this discrepancy. Secondly, the OA pathways included and the parameters used in the model still have uncertainties. Thirdly, the model resolutions are not high enough to capture the hot points at cities with small urban areas, especially the cities in western China (e.g., Lhasa and Dunhuang). Overall, the model explained most of the observations. In Fig.4b, it can be seen that SOC dominated most regions of the globe with the fraction above 70%. Over China, POC dominated the eastern regions while SOC dominated the western regions. In the eastern regions, the higher primary emissions lead to the lower fraction of SOC in OC although the SOC concentrations are higher than that in western regions.

In the VBS in our model, there are three pathways forming SOA, i.e., oxidation of POA, oxidation products of anthropogenic and biogenic VOC and IVOC. Simulations of most previous studies (Kanakidou et al., 2005; Tsigaridis et al., 2014) showed that biogenic SOA (BSOA) is dominant over the global scale due to their major sources from the oxidation of biogenic VOCs. However, our simulations shown in Fig.5 indicate that anthropogenic SOA (ASOA) are as important as the biogenic one, especially over the areas with large anthropogenic emissions. Over some areas in India and eastern China, concentrations of ASOA can be above 7 μg/m$^3$, significantly greater than the concentrations of BSOA ( $< 3$ μg/m$^3$). Even over south America and Africa, ASOA has concentrations of 1~3 μg/m$^3$ due to the large contribution of IVOC

and POA emitted from biomass burning. The higher concentrations of ASOA than BSOA are also demonstrated by other studies. For example, adding an additional SOA correlated with the CO emission can improve the observed OA concentration (Spracklen et al. 2011). In the second domain simulation (shown in Fig.5c and 5d), it is more clearly seen that ASOA has the higher concentrations than BSOA over China. In North China Plain, concentrations of ASOA were above 3 μg/m$^3$ while concentrations of BSOA were below 1 μg/m$^3$. Previous modeling studies using VBS (Han et al., 2016; Lin et al., 2016) also suggested that ASOA is dominant in North China. Observation analysis indicated ASOA was the highest one among the contributors of SOA sources, very different from the reported cases of developed countries (Ding et al., 2014; Li et al., 2017; Tang et al., 2018). In addition, our simulation considered the SOA formation from IVOC, which has been proved to be a large contributor to SOA (Zhao et al., 2016; Yang et al., 2019). Clearly, it is the allowing of POA to be volatile and including the SOA formation from IVOC that constitute the larger sources of ASOA. The substantial contribution of ASOA to SOA suggests the significant role of ASOA in particle growth over the areas with intense anthropogenic emissions, which will be discussed in Sect.5.4.

Volatility distribution of SOA is a factor controlling not only the mass concentrations of OA but also the size distributions of aerosol particles via microphysical processes. The framework of VBS can simulate the volatility distribution of OA in five saturation concentration bins. Study of Riipinen et al. (2011) suggested that roughly half of the condensing mass needs to be distributed proportional to the aerosol surface area to explain the observed aerosol particle growth. The condensation of this part of OA is governed by gas-phase concentration rather than the equilibrium vapour pressure, which is the way our model calculates the growth of LV-SOA to particles. The volatility distribution of SOA is an important factor impacting the global and regional distribution of particle number concentration. Fig. 6 shows the surface layer spatial distributions of SV-SOA and LV-SOA concentrations. Globally, the high concentrations of SV-SOA are mainly located in the continental source regions. By contrast, the distribution of LV-SOA is more

homogeneous and its contribution to SOA is lower in source regions. The continent with higher emission has a lower contribution of LV-SOA. In the downwind regions, LV-SOA has the higher concentration than SV-SOA. Even over the source areas, such as north America and Europe, LV-SOA also has the higher concentration than SV-SOA. These results indicate the multi-generation aging of OA in VBS produce the higher concentration of LV-SOA and thus the wider spread of OA, which would have a large impact on the role of OA in particle formation processes. Over China, SV-SOA has the concentration of 3-10 μg/m$^3$ and is dominant over source areas in eastern region. LV-SOA has the concentration of 2-5 μg/m$^3$ in the eastern region and 0.6-2 μg/m$^3$ in the western region. Measurement analysis suggested that OA and SOA of Beijing in China are more volatile than those of the cities in Europe and America (Xu et al., 2019). Our study indicated that, in addition to the different emission sources, the more volatile of SOA is also caused by the relative lower contribution of LV-SOA to SOA although the concentration of LV-SOA over eastern China is higher than that over Europe and America.

### 5.3 Global and regional distribution of particle number concentration

Fig.7 displays the simulated surface layer horizontal spatial distributions of annual mean number concentrations of CN10 and the fraction of CN10 that is secondary. The observed CN10 values given in Table 3 are also shown in Fig.6a and Fig.7c for comparison. In Fig.7a, it is clear that high concentrations of CN10 in the surface layer are located in the regions with large anthropogenic emissions. Highest concentrations of annual mean CN10 are over the central-eastern China and Sichuan basin and their values can be larger than 10000 cm$^{-3}$. Over eastern America, most areas of European developed countries, and India, values of annual mean CN10 are over 5000 cm$^{-3}$. Over South America and South Africa, CN10 concentrations are also higher due to the biomass burning emission. Affected by continental sources and ship emissions, CN10 concentrations over the coastal regions and adjacent seas close to the continent can be over 1000 cm$^{-3}$. Over the polar regions and the oceans far from continents, CN10 concentrations are lower than 300 cm$^{-3}$. The model well reproduced the above spatial variation of CN10 represented by observations in different

environments. By a more specific comparison in Fig.8, where the values of simulation are compared by a scatter plot with corresponding observations at 34 sites given in Table S3, the simulations of annual mean concentration of CN10 agree quite well with the observations, within a factor of two for most of the sites. The spatial pattern of CN10 over the second domain (in Fig.7c) is similar with that of the corresponding region in the first domain (in Fig.7a), but the gradients of CN10 is characterized more precisely due to the higher horizontal resolution. For example, the high concentrations of CN10 over southern Hebei are better depicted in Fig.7c than Fig.7a. The observed annual mean CN10 concentration (12000 $cm^{-3}$) at Shangdianzi in eastern China was five times greater than that of Waliguan (2030 $cm^{-3}$) in western China. The corresponding simulated CN10 concentrations, 14380 $cm^{-3}$ and 2780 $cm^{-3}$, well reflected this regional difference.

Both secondary particles formed through nucleation and subsequent growth and direct emission of primary particles can contribute to atmospheric particle number concentration. It is important to quantify the contribution of these two sources in different parts of the globe. In Fig.6b, it can be seen that secondary particles are dominant in most parts of the globe except for the regions with large primary emissions, e.g., eastern China, India, and southern Africa. The low contribution of secondary particles in these regions is due to the strong scavenging of secondary particles by primary particles and the low nucleation rate caused by competing of primary particles for condensable gases. This spatial pattern is similar with the results of previous studies (Yu and Luo, 2009). However, the fractions of secondary particles in CN10 are lower than those in CN3 showed in Yu and Luo (2009) due to the dominant contribution of secondary nucleation to particles in 3-10 nm. In Fig.7d, a boundary from northeast to southwest can be seen to separate the areas dominated by secondary particles from that by primary particles over China. This phenomenon is also caused by the large difference of emissions between western region and eastern region.

**5.4 The mixing state of organic aerosols and their growth to new particles**

Besides particle number concentration, mixing state of aerosols is necessary to

evaluate aerosol impacts on climate. The framework of VBS treats the emitted organic aerosol with volatility and thus allows them to be partitioned among different aerosol particles through condensation. In addition, the evolving volatility due to oxidation in the atmosphere makes microphysical behavior of POA to be different from the nonvolatile POA. In our model, semi-volatile organics is temperature-driven partitioned by equilibrium assumption while low-volatility species is kinetically condensed on the particles. Fig.9 shows the fraction of organic species reside in aerosols of different types (i.e., SPs, sea salt, dust, BC, and OC) defined in our model. In Fig.9, most of the organic species reside in OC, SPs, and BC particles, suggesting the intense mixing of anthropogenic aerosol species. In the southern hemisphere, the fractions of organic species residing in SPs are above 30%, larger than that of OC particles. In the northern hemisphere, organic species mainly reside in OC particles due to the higher concentration of POA and the subsequent partition. The fractions of organic species reside in SPs are lower, but still considerable, indicating the important role of organic species in forming particles over the whole globe. Due to the different emission and the associated microphysical processes, there are distinct spatial variations of organic species distribution among different continents. Over America, 30%~40% of OA resides in SPs. By contrast，this fraction is below 20 % over China. In China, significant difference also exists between the western and eastern region. The dominant contribution of semi-volatile species to OA (shown in Fig.6) and their partition proportional to the low-volatility OA lead to a higher fraction of organic species residing in OC particles over eastern China. The mixing of natural aerosols and organic species were also demonstrated in Fig.9. Over the most areas of the globe, 15% of organic species are distributed in dust particles, which could greatly modify the properties of dust particles and thus their climate forcing over the regions influenced by dust particles (Huang et al., 2019).

Previous study indicate that organic species are the major components of aerosols (e.g., Zhang et al., 2007; Jimenez et al., 2009) and low-volatility organic species can greatly enhance the growth of new particles (e.g., Yu, 2011; Tröstl et al., 2016). Our results presented above also indicated the substantial distribution of

organic species in SPs. For this reason, the contribution of LV-SOG to the growth of SPs is analyzed. Fig.10 shows the ratio of LV-SOG to $H_2SO_4$ and the ratio of low-volatility organic species to sulfate that reside in SPs. The concentration of LV-SOG is a factor of ~1.5-10 higher than that of $H_2SO_4$ over many parts of the continents and the adjacent oceans but is lower in East Asia, eastern United States, southern Europe, and northern Africa where emissions of $SO_2$ are high. Especially, over the areas in Sichuan Basin and eastern China (shown in Fig.10c), the concentrations of $H_2SO_4$ are significantly higher than that of LV-SOG. Compared with the simulation of Yu (2011), our results included anthropogenic LV-SOG and therefore the ratios of LV-SOG to $H_2SO_4$ are higher, especially in the regions influenced by continental sources and oceans with ship emissions. In Fig.10b and Fig.10d, the contribution of low-volatility organic species to the growth of SPs, presented by the concentration ratio of low-volatility organic species to sulfate, is higher in the southern hemisphere and lower in the northern hemisphere where continental sources of $SO_2$ are larger. Though lower, the contribution is considerable (~10-20%) over Europe and north America. Similar with the contribution of ASOA to SOA, LV-SOA residing in SPs is dominated by anthropogenic ones over POA source areas (as the case in Beijing shown in Fig.3). The condensation growth of SPs by low-volatility organic species can enhance their survival rate and therefore could increase the contribution of SPs to particle number concentration. These results highlight the importance of ASOA in new particle growth over the polluted regions, such as eastern China and India.

**5.5 Sensitivity of particle number concentration to volatility of POA**

In the VBS, POA is treated as volatile species and allowed to be aged by oxidation in the atmosphere, it is necessary to explore the uncertainties associated with this treatment of volatility distribution. In addition, the size distribution of POA and the associated microphysical processes are also modified due to this treatment. For this reason, the sensitivity of particle number concentration to the volatility of POA and the assumed size distribution of PPs are discussed here. Fig.11 displays the change ratio of number concentrations of CN10 in LV_POA experiment, HV_POA

experiment, PPD0.5 experiment, and OCD0.5 experiment to that in BASE experiment. Overall, concentration of CN10 changed a little when POA volatilities were in the inter-quartile range of measurements (shown in Fig.11a and b). When using the low/high volatility distribution of POA, PPs number concentrations were increased/decreased by 5-10% over the most areas in the northern hemisphere. By contrast, SPs number concentrations only had a little change over the areas with the strongest emissions. Due to the dominant contribution of SPs, CN10 had no clear change in most regions of the globe. By contrast, the size distribution of emitted PPs has large influence on the concentration of CN10. When the median diameter of BC and OC use the half size of the BASE experiment, concentrations of CN10 were increased by 50-150% over the areas with large emission sources of BC and OC, which was too high to matching the observations shown in Fig.7a. For example, the CN10 in PPD0.5 experiment were greatly overestimated when compared with observed concentration (12000 $cm^{-3}$) at Shangdianzi in eastern China. Therefore, halving the median diameter of BC and OC in the PPD0.5 experiment could not represent the real situation. However, halving the median diameter of OC only leads to the increase of CN10 by 10-50% over eastern China. Over the other areas with high emission, there was no observation available for comparison. Considering the other factors affecting the simulation of CN10, it is not safe to say that the assumed median size of OC is too small in the PPD0.5 experiment. Moreover, the emitted OC particles indeed have volatility and can re-evaporation after dilution (Robinson et al., 2007; Donahue et al., 2009). The assumption of OC size distribution should take volatility of OC into account. It is necessary to measure the size distribution of freshly emitted primary particles and compare the model results with observation in polluted atmosphere dominated by PPs to clarify this issue.

## 6 Conclusion and discussion

The sources of organic aerosol, volatility distribution, and the microphysical behavior of organic species have been found to be important in particle formation processes by laboratory studies and field observations, but the organic aerosol processes are still poorly represented and they are the large contributors to model

uncertainties in simulating aerosol microphysical properties. In this study, a new global-regional nested aerosol model was developed to simulate detailed microphysical processes in the real atmosphere. The new model combined APM module and VBS module to simulate microphysical processes of OA. In the model, the OA in the lowest volatility bin is treated as non-volatile/low-volatile species and their condensation was simulated by the kinetic way. The OA in other volatility bins was simulated by equilibrium partitioning. Using this framework, both the condensation of secondary inorganic species (i.e., sulfuric acid, nitrate, and ammonium) and the condensation of organic species with different volatilities (i.e., low-volatility and semi-volatile organic compounds) were simultaneously simulated, which is an important advances of our new model. The concentration of low-volatility organics is separately calculated and the condensation of $H_2SO_4$ and LV-SOG on size-resolved secondary particles is explicitly simulated, along with the condensation of LV-SOG on primary particles. Therefore, the growth of LV-SOG to new particles and aging of primary particles by organic species were represented in a realistic way. Compared with the most models in the second phase AeroCom (Tsigaridis et al., 2014; Mann et al., 2014) and the recently developed new models (e.g., Yu, 2011; Patoulias et al., 2015; Gao et al., 2017), our model includes the more comprehensive sources of SOA by using the VBS framework, especially the anthropogenic SOA. In addition, allowing POA to evaporate and re-condense onto the particles make its microphysical behavior more like SOA and therefore give new meaning to the POA–SOA split which significantly affects the global CCN formation (Trivitayanurak and Adams, 2014). The flexible framework of APM combined with VBS produces the different distribution of organic species in aerosols, i.e., the mixing state of OA, which has been found to cause substantial difference in radiative effects of aerosols (Zhu et al., 2017). Box model analyses showed that the low-volatility SOA has a large fraction in the growing nucleation mode particles (Pierce et al., 2011). The comprehensive thermodynamic-kinetic approach treating the condensation and the partitioning of organic species originated from biogenic and anthropogenic sources allows us to investigate the full role of organic species in the growth of new particles, which is

important for understanding the formation processes of particles relevant for radiative forcing and clouds (Shrivastava et al., 2017).

The model with three nested domains was applied to simulate the aerosol components and PNSD in Megacity Beijing during a period of about a month. The simulation results were evaluated by the observations at the high level of IAP tower, which is more representative than the ground level in regional scale. The simulated BC and OA components agreed well with the PMF analysis of AMS measurements. The evolution of PNSD and NPF events were also nicely reproduced by the model. Our modeling analyses showed that AOA accounts for the larger part of OA of SPs and thus significantly contributed to the growth of SPs in Beijing. Molteni et al. (2018) indicated highly oxygenated organic compounds formed from anthropogenic VOCs can substantially contribute to NPF in urban areas. Observations in Beijing suggested that anthropogenic VOCs are major constitutes of SOA (Ding et al., 2015; Yang et al., 2016). For the first time, the contribution of AOA to new particles was quantified benefiting from the mixing state our model resolved in our study. Although the exact role of AOA in NPF is not quite clear, our study explicitly displayed the important role of AOA in NPF in Chinese megacities, which would help elucidate the mechanism of more frequent occurrences of NPF events than theoretical prediction in polluted atmosphere (Kulmala et al., 2017; Chu et al., 2019). By comparison with the observations collected from published data, the model well reproduced the annual mean concentration of the observed OC at continent sites in America and China. Due to the re-evaporation and oxidation of POA and the additional emission of IVOC, ASOA becomes dominant in SOA over POA source areas. At sites in different environments over the globe, the model produced the reasonable concentrations of CN10 within a factor of two for most of the sites. LV-SOG, especially the anthropogenic SOA, was found to have a large contribution to new particle growth over areas with intense anthropogenic emissions, such as in eastern China. The global simulation of Kelly et al. (2018) found that including the large anthropogenic SOA source could get results consistent with observations over the northern hemisphere mid-latitudes. Simulation over East Asia also indicated that most of the OA were from

anthropogenic sources (Matsui et al., 2014). Together with these studies, our modeling results further provided the direct evidence of AOA in particle formation processes not only in Chinese megacities but also in other regions influenced by anthropogenic sources in the global scale.

Sensitivity analyses indicate that concentration of CN10 only changed a little in the regions with the highest emission of POA and had no clear change in most regions of the globe when POA volatilities were in the inter-quartile range of measurements. Although the size distribution of primary emitted particles has large impact on the simulation of CN10 as suggested by other studies (e.g., Spracklen et al., 2006; Chang et al., 2009; Zhou et al., 2018), the simulation of the base experiment gave the better agreement with the observations than the sensitivity experiments and the conclusions will not be changed. Even so, the importance of the size distribution of primary emitted particles should be emphasized. The global model results have suggested the high sensitivity of CCN to to the assumed emission size distribution (Lee et al., 2013). Recently, Xausa et al. (2018) found that using the size-segregated primary particle number emissions could make the number concentration of accumulation mode particles closer to the measurements. Here, our simulation indicates to the importance of parameterization of the size distribution of emitted OC particles after considering their re-evaporation and condensation. Therefore, it is necessary to constrain the primary emission both in their size distribution and volatility. In addition, it should be noted that the simulated properties of OA were also determined by the parameters of VBS module, the emissions inventory and meteorological fields input to the model, and the physicochemical processes in the model. Although our model gave the reasonable calculation comparable with the available observations and model results of other authors, it is still necessary to further improve our model in the future. For example, the size-resolved emissions of anthropogenic primary particles will be used as the model input to reducing the uncertainties associated. More nucleation schemes will be implemented into the model to investigate the influence of nucleation schemes on the aerosol number concentrations as the uncertainties from nucleation scheme are still large (Dunne et al., 2016). Aqueous-phase formation processes of SOA have

evident influence on the particle properties and total SOA mass (Ervens et al., 2011) and these processes can close the gap between the simulation and observation (Lin et al., 2014). It is necessary to refine the description of aerosol microphysical processes by including aqueous formation of SOA in our model.

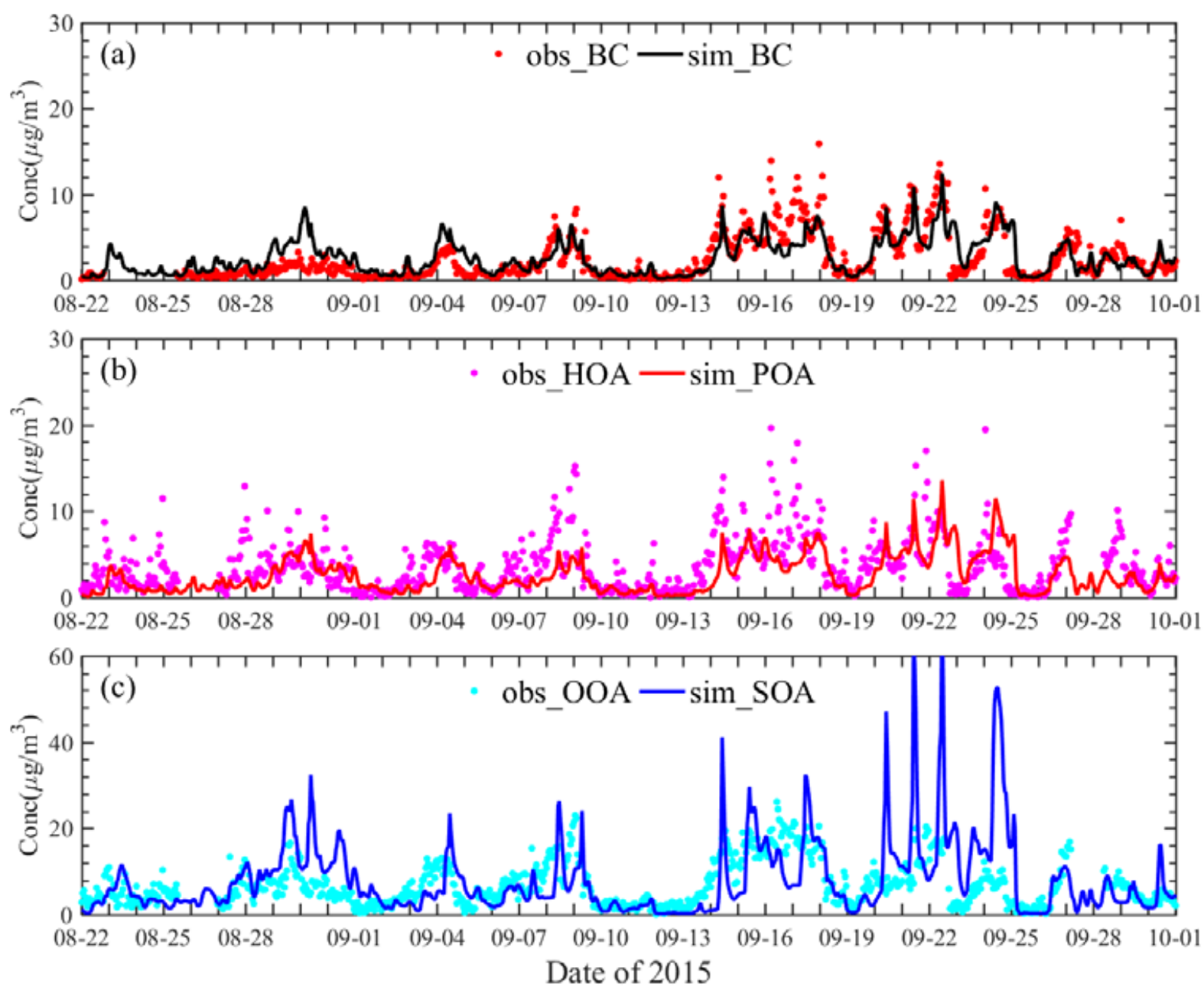

Fig.1 Comparison of the simulated and the observed (a) black carbon, (b) primary organic aerosol, and (c) secondary organic aerosol at the 260m height in Beijing from August 22 to September 30, 2015. All the observations were shown with dot points and the simulations with lines.

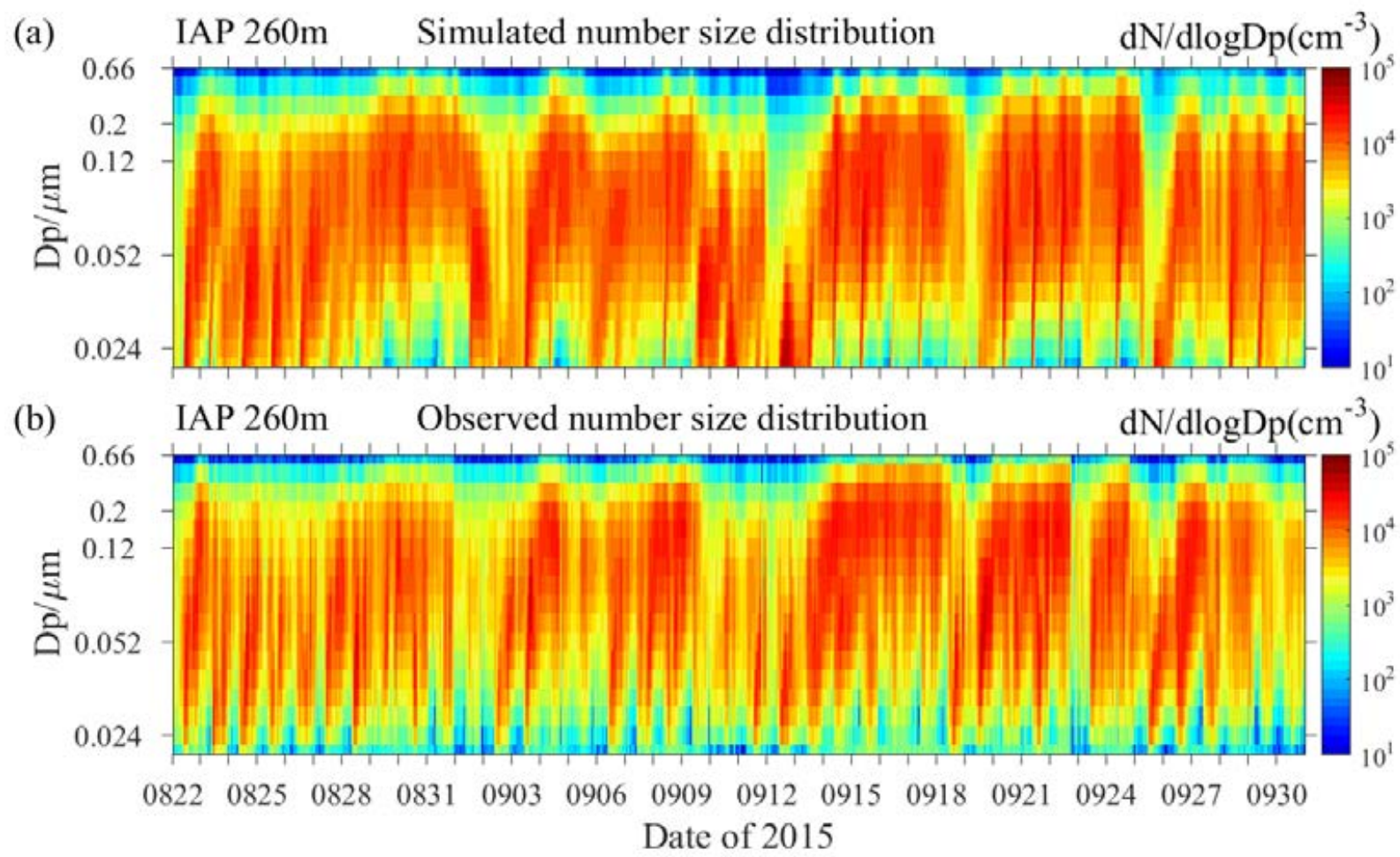

Fig.2 (a) Simulated and (b) observed particle number size distribution at high level (260 m) in Beijing from August 22 to September 30, 2015.

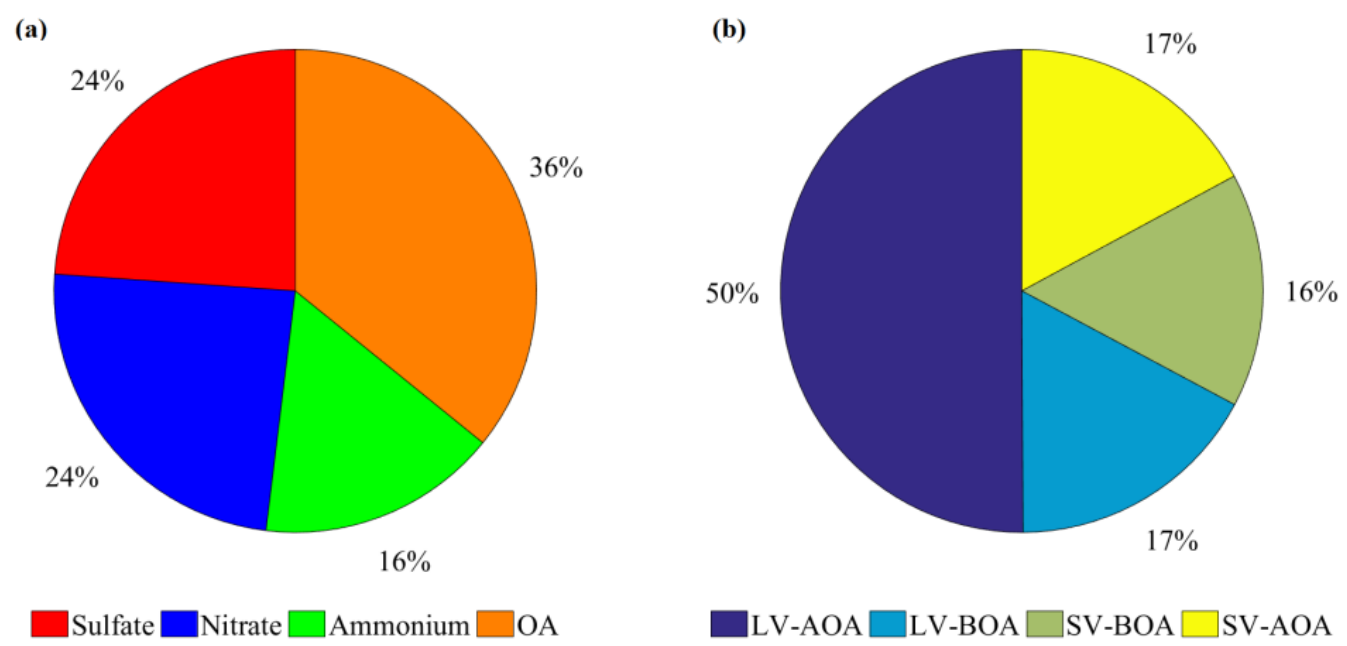

800 Fig.3 (a)The mean contribution of sulfate, nitrate, ammonium, and OA to the mass concentration of SPs and (b) the mean contribution of LV-AOA, LV-BOA, SV-BOA, and SV-AOA to the mass concentration of OA in SPs in September, 2015.

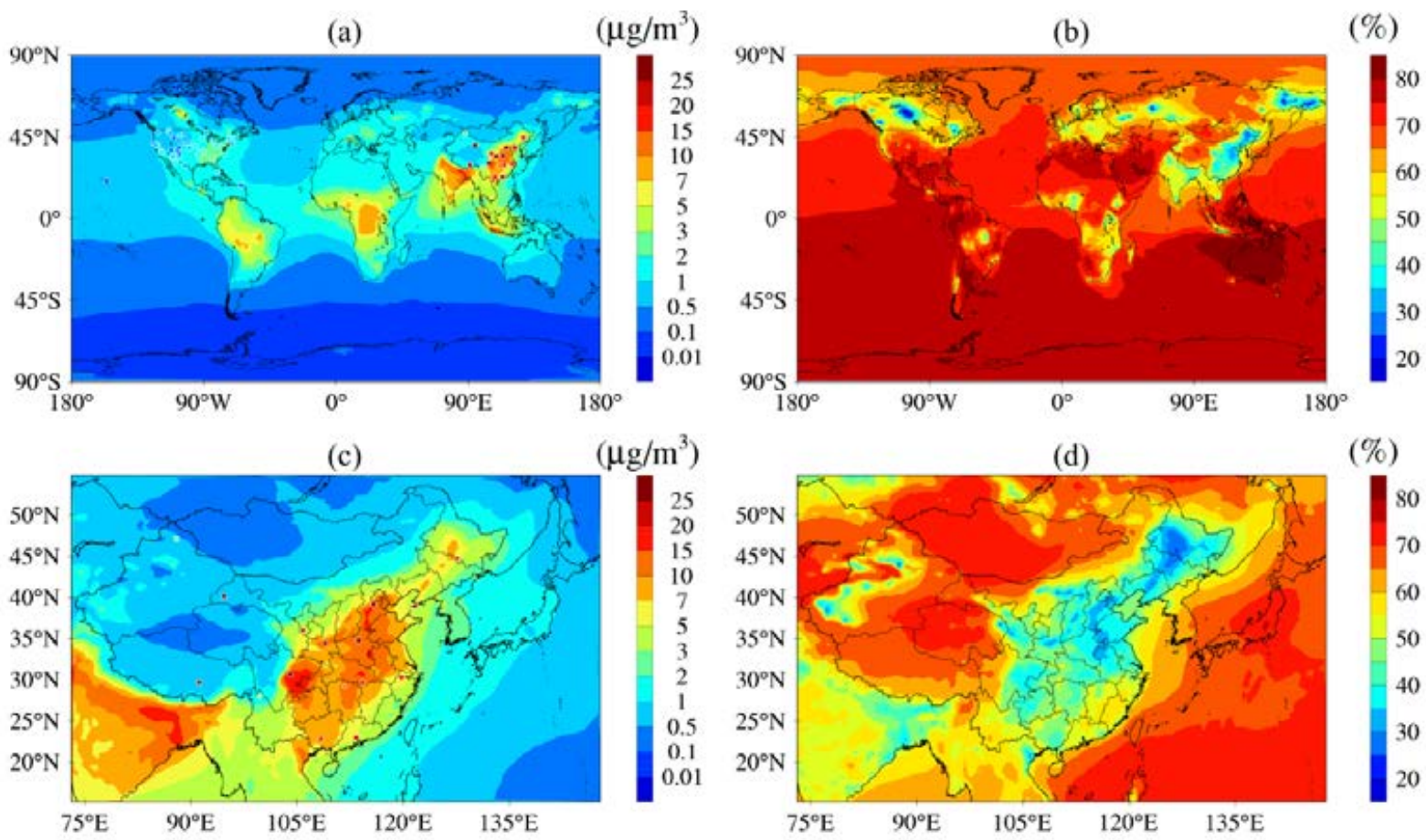

Fig. 4. Surface layer horizontal spatial distributions of organic carbon concentrations (left panel) and the fraction of OC that is secondary (right panel) over the first domain (top panel) and second domain (bottom panel). Observed OC collected in Sect.4 are also overlapped with shaded circles on the plots for comparison.

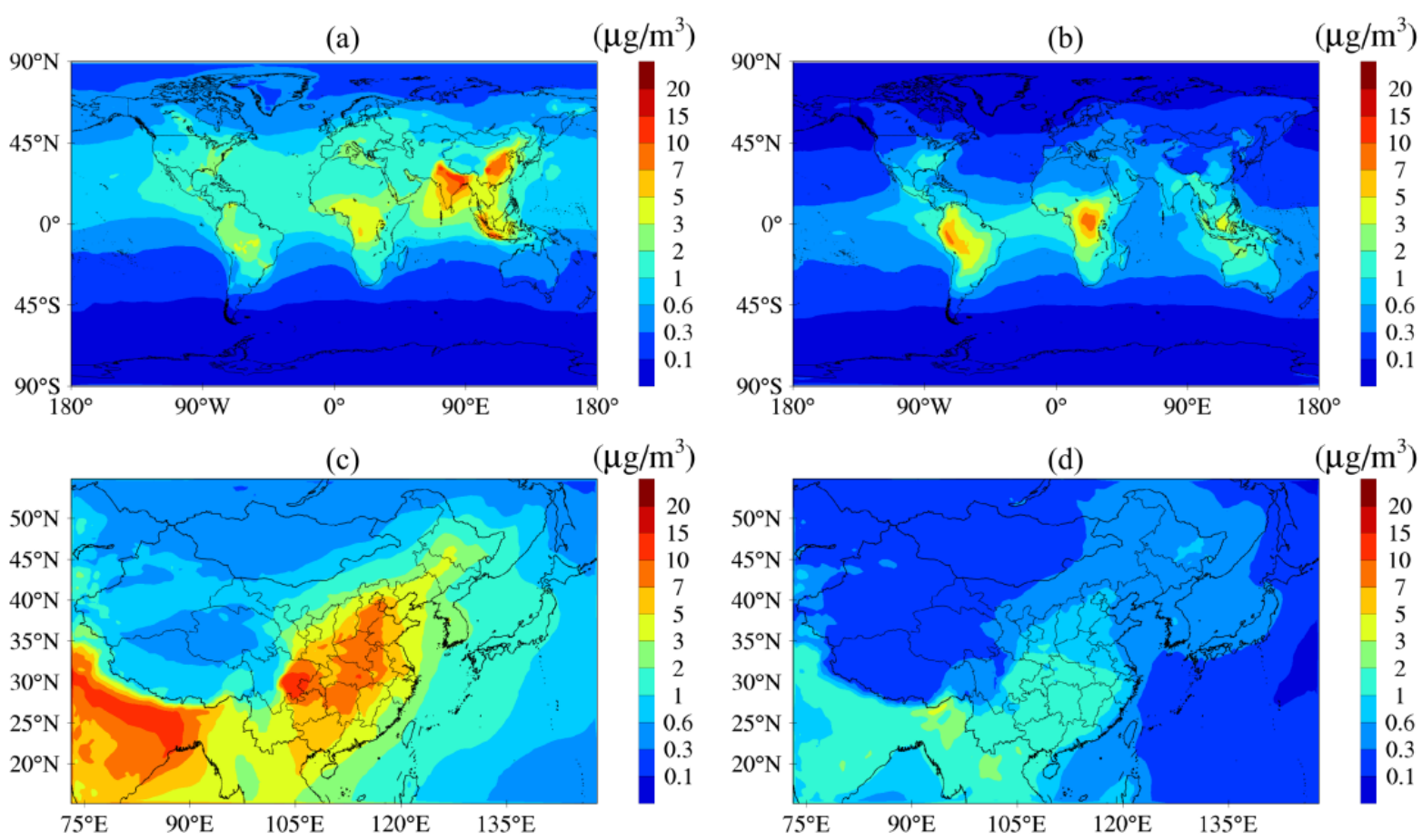

Fig. 5. Surface layer horizontal spatial distributions of ASOA concentrations (left panel) and BSOA concentrations (right panel) over the first domain (top panel) and second domain (bottom panel).

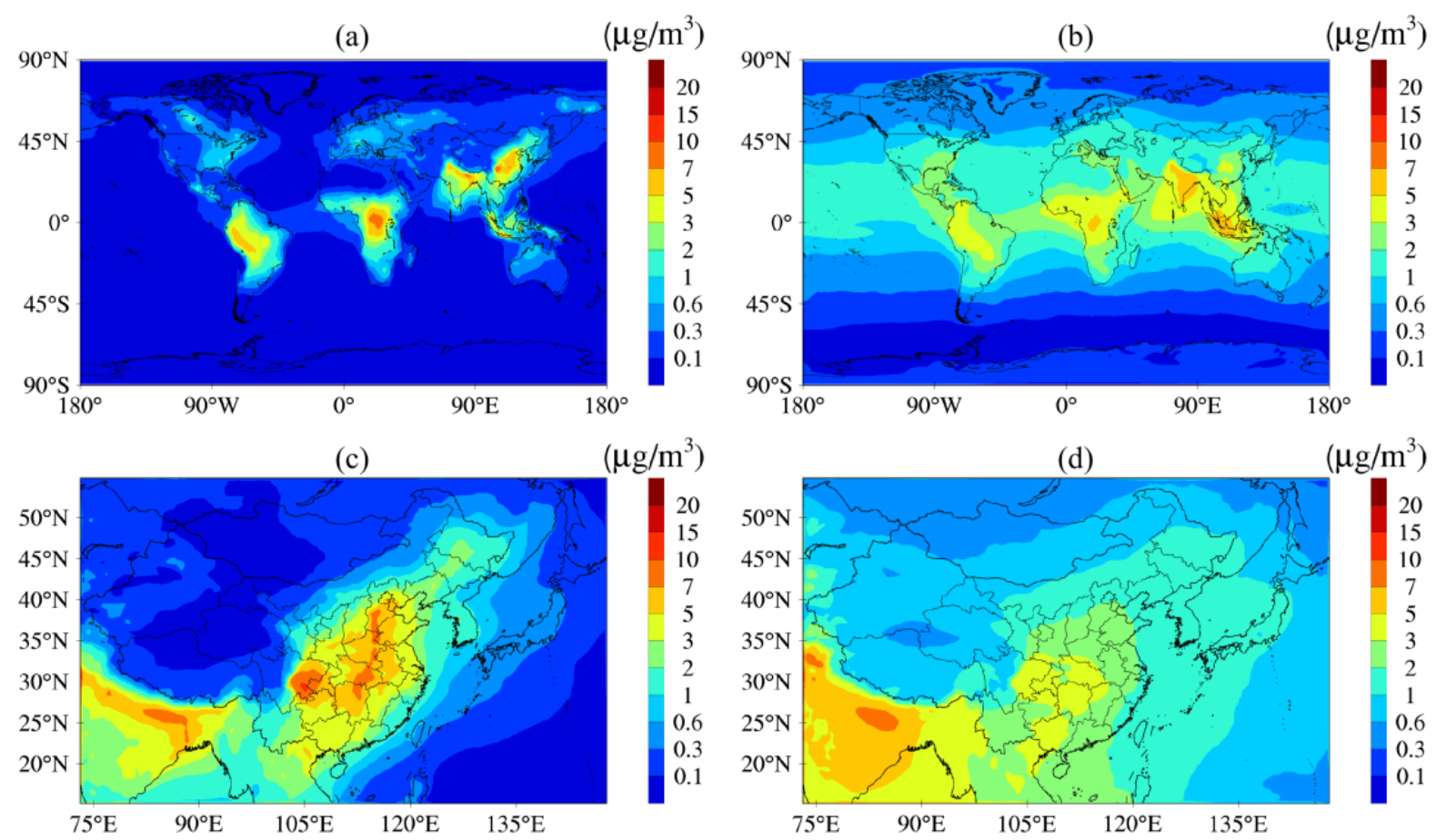

Fig. 6. Surface layer horizontal spatial distributions of SV-SOA concentrations (left panel) and LV-SOA concentrations (right panel) over the first domain (top panel) and second domain (bottom panel).

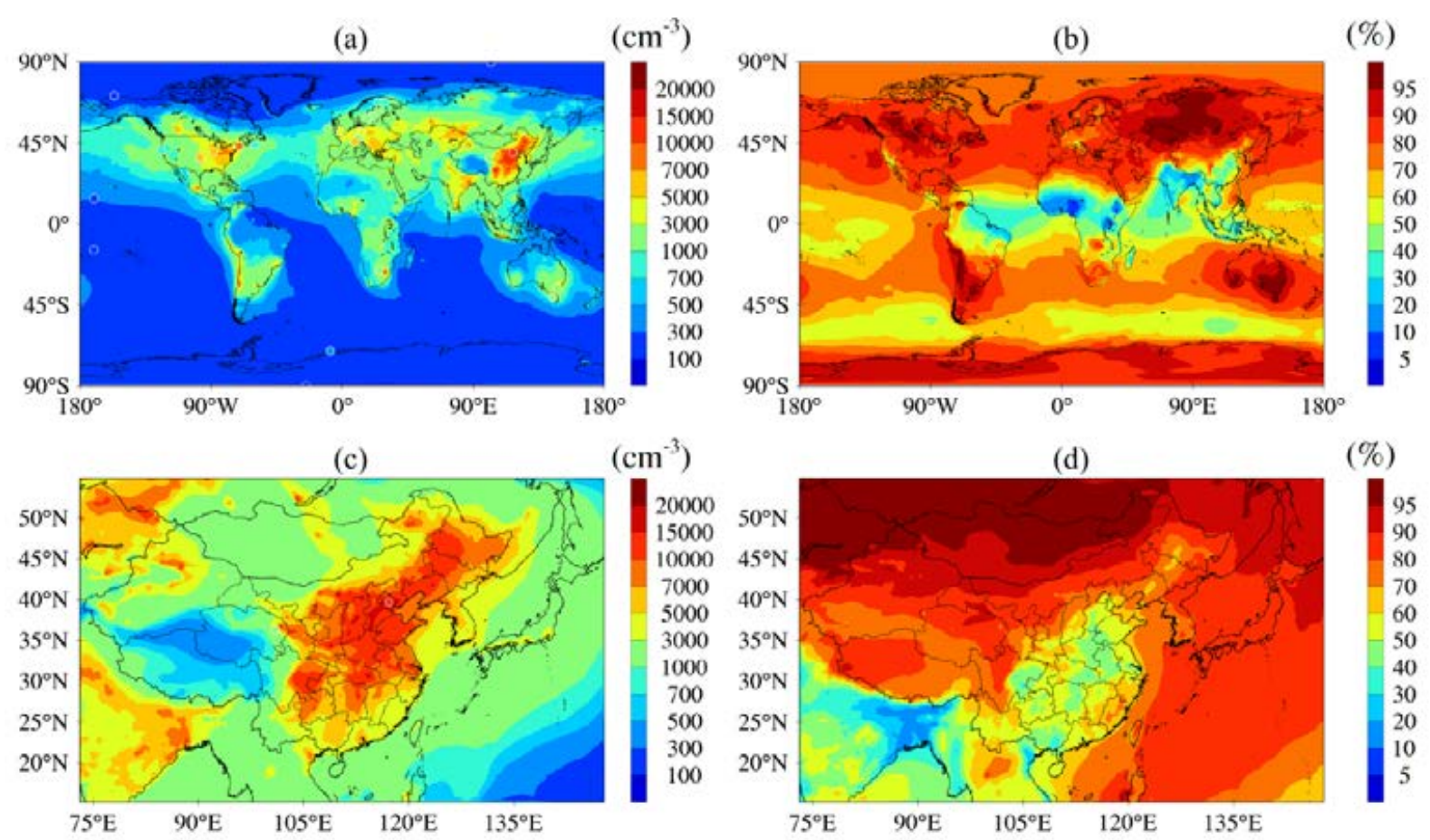

Fig. 7. Surface layer horizontal spatial distributions of annual mean number concentrations of CN10 (left panel) and fraction of CN10 that is secondary (right panel) over the first domain (top panel) and second domain (bottom panel). Observed CN10 values in Table 3 are also overlapped with shaded circles on the plots for comparison.

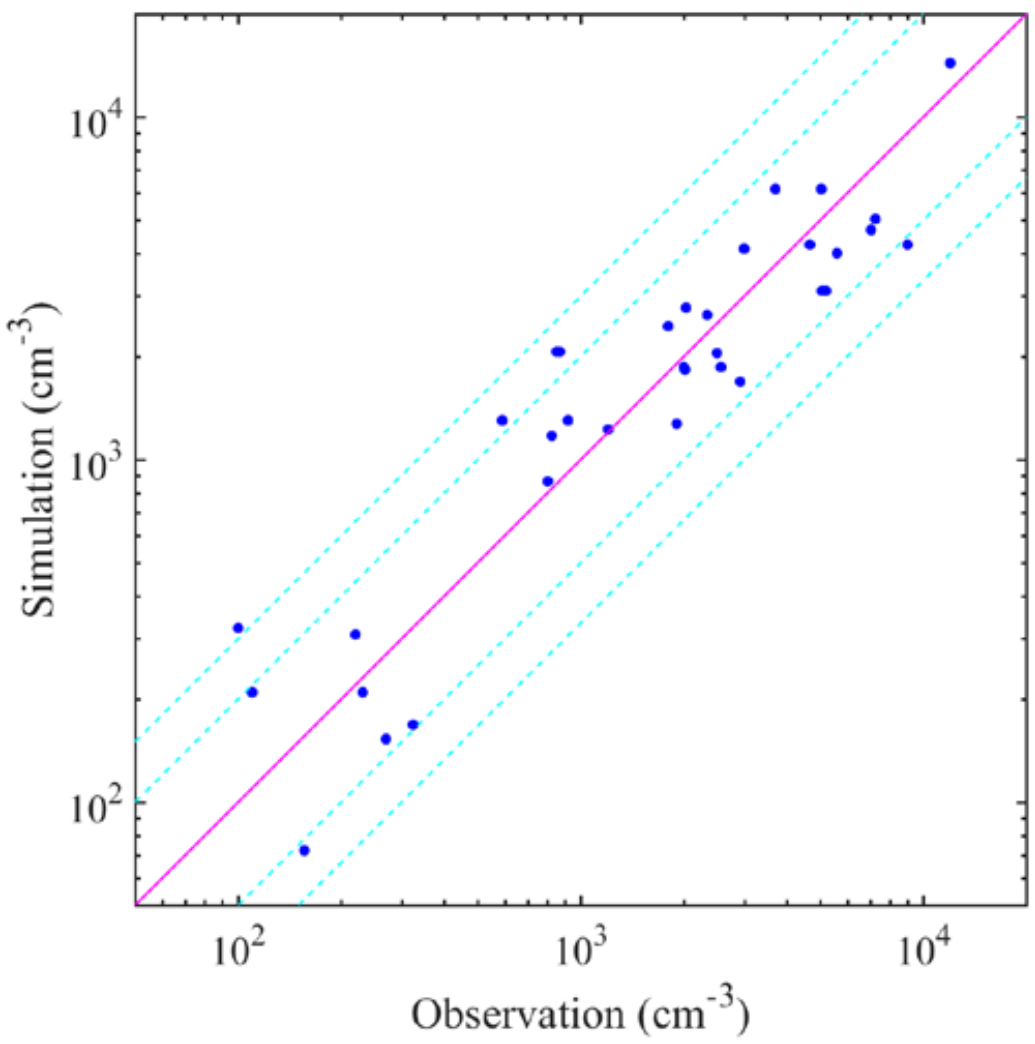

Fig. 8. Comparison of simulated and observed annual mean number concentrations of particles condensation larger than 10 nm at 34 sites listed in Table 3. The solid carmine line shows a 1:1
830 ratio and the dashed turquoise lines show ratios of 3:1, 2:1, 1:2, and 1:3.

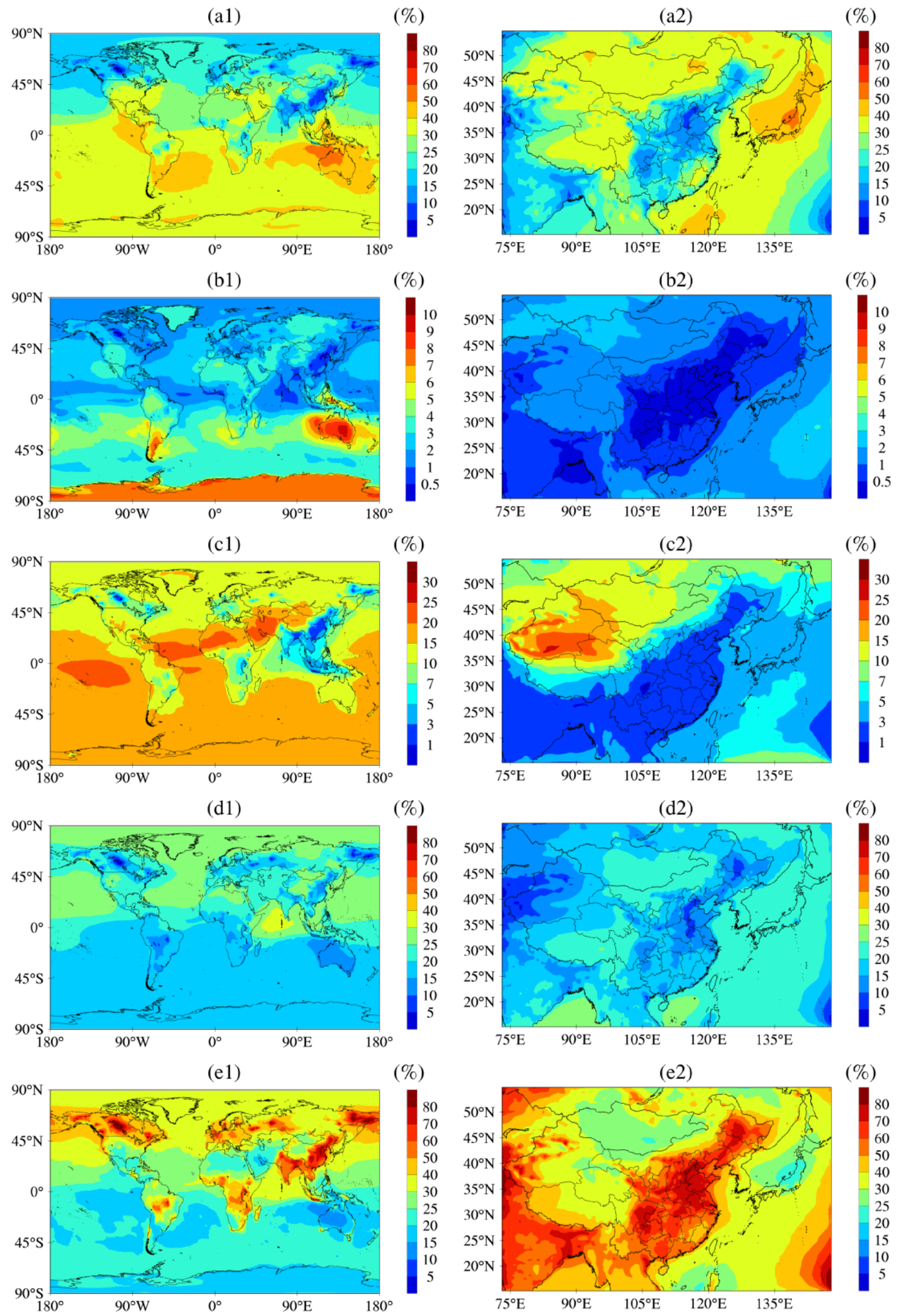

Fig. 9. Surface layer horizontal spatial distributions of the fraction of organic species that reside in SP (a1 and a2), sea salt (b1 and b2), dust (c1 and c2), BC (d1 and d2), and OC (e1 and e2) particles over the first domain (top panel) and second domain (bottom panel).

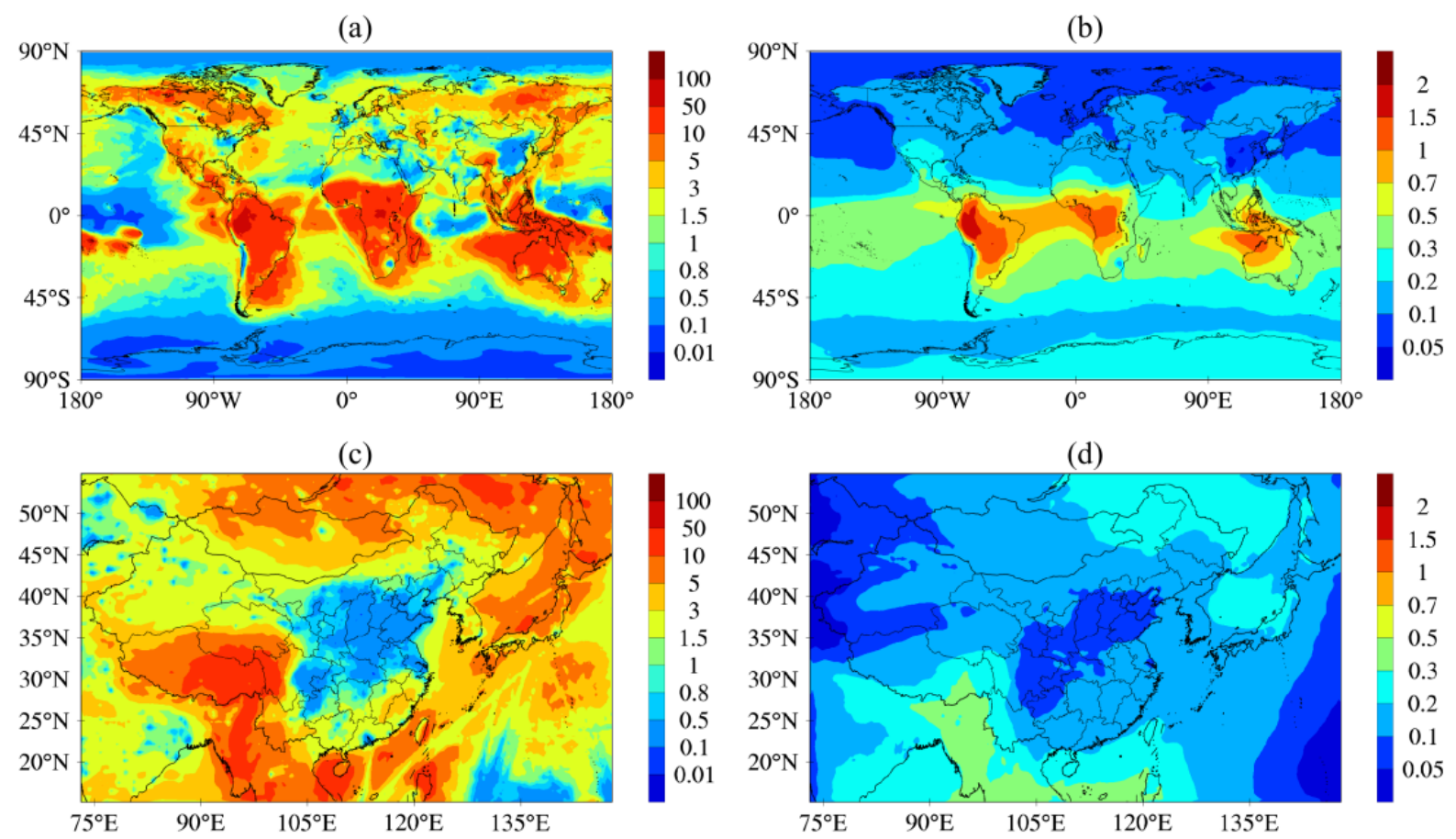

Fig. 10. The ratio of LV-SOG to $H_2SO_4$ (left panel) and the ratio of LV-OA to sulfate (right panel) that reside in SPs. Top panel is for the first domain and bottom panel for the second domain.

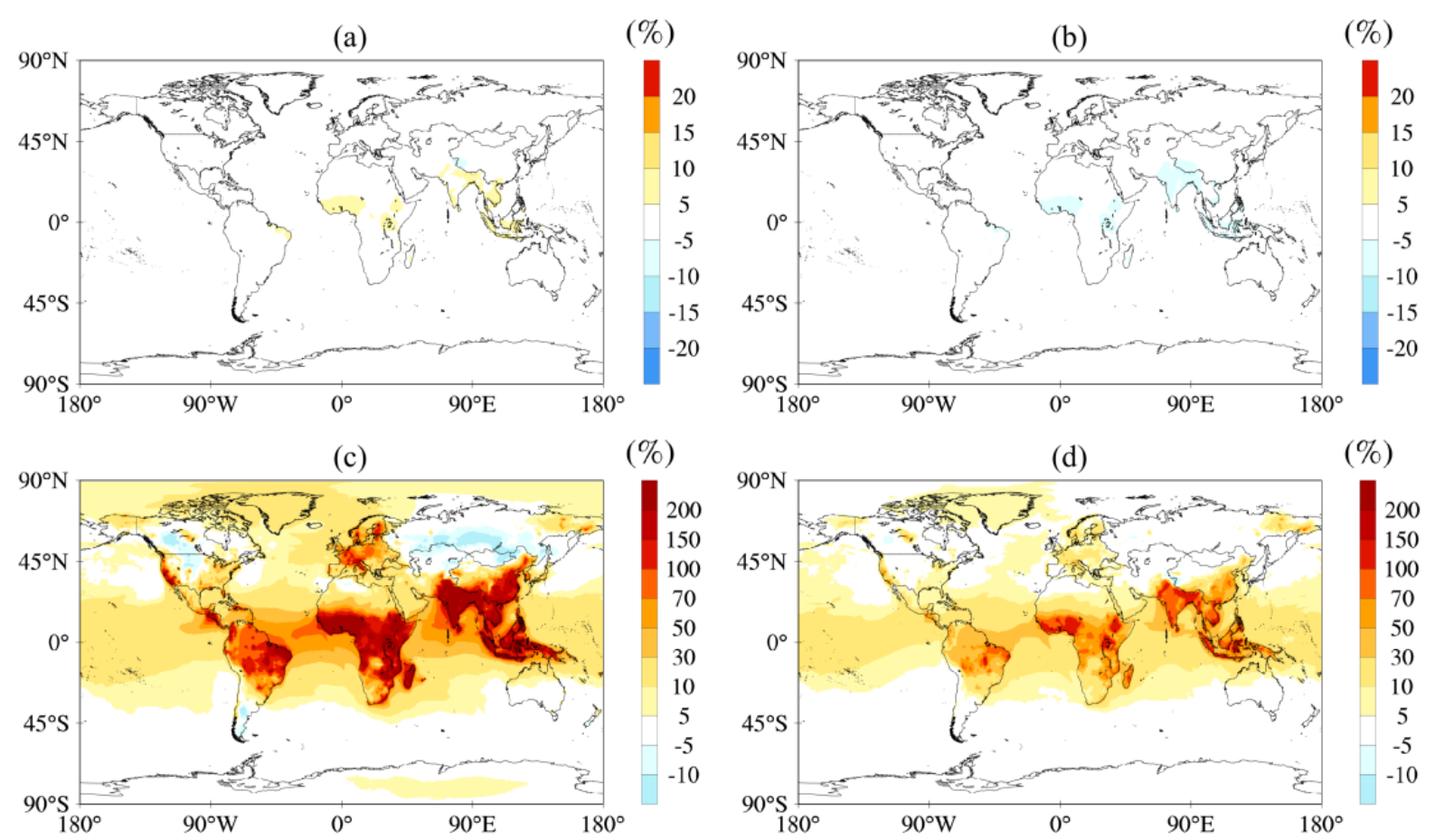

Fig. 11. Relative change of number concentrations of CN10 in (a) LV_POA experiment, (b) HV_POA experiment, (c) PPD0.5 experiment, and (d) OCD0.5 experiment to that in BASE experiment.

Table 1 The newly added tracers for simulation in microphysical processes

| Tracers | Description |
|---|---|
| $H_2SO_4$ | Sulfuric acid gas |
| LV-SOG | Low-volatility secondary organic gas |

| | |
|---|---|
| Sulfate(1-40) | Size-resolved sulfate of secondary particles in 40 bins |
| BC(1-28) | Size-resolved black carbon in 28 bins |
| POC(1-28) | Size-resolved primary organic carbon in 28 bins |
| Sea salt(1-20) | Size-resolved sea salt in 20 bins |
| Dust(1-4) | Size-resolved dust in 4 bins |
| BC_Sulfate | Sulfate coated on BC |
| OC_Sulfate | Sulfate coated on OC |
| Sea salt_Sulfate | Sulfate coated on sea salt |
| Dust_Sulfate | Sulfate coated on dust |
| SP-LV | Low-volatility organic aerosol coated on SPs |
| Salt-LV | Low-volatility organic aerosol coated on sea salt |
| Dust-LV | Low-volatility organic aerosol coated on dust |
| BC-LV | Low-volatility organic aerosol coated on BC |
| OC-LV | Low-volatility organic aerosol coated on POC |

Table 2 Sensitivity experiments and their description. The “Primary size” column refers to the geometric mean diameter values (nm) assumed for primary carbonaceous aerosol emissions. The "Volatility distribution" column refers to the coefficients of POA distributed to the volatility bins for vehicles, other anthropogenic, biomass burning, respectively. Coefficients for different sources are separated by semicolons, and different bins (from the lowest to the highest) by commas.

| Experiments | Primary size | Volatility distribution |
|---|---|---|
| BASE | 60, 150 (1.80, 1.80) for BC and POC | 0.27, 0.15, 0.26, 0.15, 0.17; 0.167, 0.167, 0.243, 0.197, 0.226; 0.2, 0.1, 0.1, 0.2, 0.4 |
| LV_POA | 60, 150 (1.80, 1.80) for BC and POC | 0.34, 0.21, 0.3, 0.1, 0.05; 0.234, 0.217, 0.27, 0.157, 0.122 ; 0.25, 0.15, 0.15, 0.2, 0.25 |
| HV_POA | 60, 150 (1.80, 1.80) for BC and POC | 0.16, 0.21, 0.21, 0.19, 0.33; 0.11, 0.093, 0.217, 0.217, 0.363 ; 0.15, 0.05, 0.05, 0.2, 0.55 |
| OCD0.5 | 30,75 (1.80, 1.80) for POC | 0.27, 0.15, 0.26, 0.15, 0.17; 0.167, 0.167, 0.243, 0.197, 0.226; 0.2, 0.1, 0.1, 0.2, 0.4 |

| | | |
|---|---|---|
| PPD0.5 | 30,75 (1.80, 1.80) for BC and POC | 0.27, 0.15, 0.26, 0.15, 0.17; 0.167, 0.167, 0.243, 0.197, 0.226; 0.2, 0.1, 0.1, 0.2, 0.4 |

***Data availability.*** All of the observation in this paper are provided in the manuscript. The simulation data can be available from the authors upon request (chenxsh@mail.iap.ac.cn, zifawang@mail.iap.ac.cn).

***Author contribution***. XC developed the model, performed the simulations and analysis, and prepared the manuscript with contributions from all co-authors. FY provided the code of APM module and modified the manuscript. WY coupled the VBS module and modified the manuscript. YS, WD, and JZ provided the observation data at Beijing site and modified the manuscript. HC prepared the emission data and modified the model code. YW, LW, HD, ZW, QW, JL, and JA modified the manuscript. ZW guided the study and modified the manuscript.

***Competing interests.*** The authors declare that they have no conflict of interest.

***Acknowledgement.*** This work was supported by the National Key R&D Program of China (Grant No. 2017YFC0209801 and Grant No. 2017YFC0209805) and the National Natural Science Foundation of China (Grant No. 41705108, 41907200, and 41907201).

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
