# Peer review of "Global-regional nested simulation of particle number concentration by combing microphysical processes with an evolving organic aerosol module"

_Atmospheric Chemistry and Physics, 2020_

## Referee Comment (RC1) · Anonymous Referee #1 · 22 Oct 2020

The microphysical processes of organic aerosol have not yet been well represented, which lead to large uncertainties in current simulation studies. In this article, authors used a new global-regional nested aerosol model combined a particle microphysics module and a volatility basis-set organic aerosol module to simulate microphysical processes of organic aerosol. The model can reproduce the organic aerosol components and the particle number size distribution in Beijing, and spatial distributions of organic carbon and number concentrations of particles condensation larger than 10 nm. They further explored the model's sensitivity to the size distribution of primary emission and

volatility distribution of primary organic aerosol. I am glad to see the amount of work presented in the manuscript. This reviewer doesn't find apparent flaw in the method and data the manuscript shows. I think the manuscript can be accepted after the following concerns are addressed. General Comments: 1. The writing is a bit difficult to understanding in many places, which leaves itself open to misinterpretations or confusion, and so the paper could really use a thorough edit from a native English speaker. 2. To provide a reliable foundation for further analysis, a comprehensive model evaluation including aerosol optical depth, PM2.5 is recommended.

Special Comments: 1. Line 85, "also" may be deleted. 2. Line 92-93, "the complication of processes and the mechanisms not well understood" may be replaced with "the unclear complication of processes and the mechanisms". 3. Line 96, "find" should be "found". 4. Line 158,"indicate" should be "indicated". 5. Line 266, "primary OA (POA)" should be "POA". 6. Line 306, "When necessary, SP-LV is redistributed to size-bins ...", please clarify the specific situations. 7. Please provide the full name for "LV-OA" and "POC" at the first appearance. 8. Line 378, "More details on the observation can be found in the published paper (Du et al., 2017)" may be "More details on the observation can be found in Du et al.(2017)". 9. Please also provide the correlation coefficients between the observed and simulated BC and simulated POA and observed HOA. 10. For figure 4 and 7, the shaded circles are difficult to observe. And the concentrations of secondary organic aerosol and CN10 are recommended to display. 11. Line 515-516, "The higher concentrations of ASOA than BSOA are also demonstrated by other studies", please some references there. 12. Please give some potential reasons for the differences between spatial distributions of SV-SOA and LV-SOA. 13. Line 563, the authors did not provide "Table 3" and "observed values in Fig.6a". Please modify. 14. Figure 7d shows that the high value CN10 is mainly primary over Northeast China where the concentration of secondary organic aerosol is relative high shown in Figure 6. Please explain this phenomenon. 15. Line 631, "indicate" should be "indicated". 16. Line 835, "top panel" and "bottom panel" should be "left panel" and "right panel".

---

## Referee Comment (RC2) · Anonymous Referee #2 · 27 Oct 2020

This manuscript presented a significant effort to couple OA formation pathways and microphysical processes to a global and regional chemical transport model, with the goal of simulating the impacts of OA physics/chemistry to particle number size distribution and mixed particle composition. The work coupled a 1.5-D VBS module and the APM microphysics module to the IAP-AACM chemical transport module. The authors also presented some preliminary comparisons with the observations, and overall the model appeared able to capture the global OA concentrations and the CN10 concentrations. The amount of work done was impressive, and the methods were mostly valid and

up-to-date.

Overall, I think the paper may be published after clarifying some missing details and improving the figure representation.

Section 2.1 Host model: What meteorological data is the IAP-AACM driven by? I see in lines 342 to 350 that the model was driven by meteorological parameters from WRF, but maybe that information can be moved here. Also, how were global meteorological fields obtained from WRF? Did the authors run Global WRF? If so, then additional references for Global WRF should be included, e.g., Zhang et al. (2012). What was the spatial resolution of the meteorological fields, and was interpolation used? What was the temporal resolution of meteorological fields? I.e., how often were the meteorological fields updated. How was the nudging performed in the three nested domains?

Section 2.3 VBS module: I would like to see the model's representation of the relationship between oxidation state and volatility expressed more clearly. Did the authors simply move the oxidation products of POA and IVOC into a volatility bin that is one magnitude lower? What about the fragmented products during the oxidation, i.e., the smaller molecular weight products?

Section 2.3 VBS module: Also, a recent paper (Jo et al., 2019) indicated that the VBS representation of SOA formation from isoprene is incorrect because the reactive uptake pathway dominates SOA formation from isoprene. Please discuss this point, the lack of reactive update pathways in this model, and the implication for the present model results.

Lines 359-360: "In the LV_POA and HV_POA experiment, quartiles of the above-mentioned distribution factors are used". Not sure what this meant. Looking at Table 2, I do not see the use of 'quartiles of the above-mentioned factors'. What different factors were used in the LV_POA and HV_POA experiments, respectively?

Table 2 and related text on the design of the sensitivity experiment: overall, I think the

sensitivity experiment could be explained more clearly. I was not able to understand what was the goals of the sensitivity experiments and how those goals relate to the parameters in Table 2.

Lines 382- 383: "...the China Atmosphere Watch Network... Zhang et al. (2008)": What year(s) were the measurements? The writing of this sentence seemed to suggest that the measurements were from 2010, which cannot be possible.

Lines 437 to 439: "The number concentration of particles from 100 nm to 1000 nm ...correlation coefficient being 0.70.': Figure 2 uses a different unit for particle size (micrometer), and I do not see the diameter extending to 1000 nm. Also, how were the normalized bias and the correlation coefficient calculated? Did the authors calculated only the bias and correlation for the time series of the total number concentration (which is not shown)? Or did they calculated a mean bias and correlation for the entire PNSD spectrum? If the latter, how was this done, and did the statistics entail a preferential weighting of the smaller particles?

Figure 4: The measurements were too small and unreadable in this figure. Please enlarge and circle with a black or white outline.

Figure 4: Also, the OC measurements in China all appeared to be much, much higher than the simulated concentrations. This is inconsistent with what was shown in Figure 1, where the authors indicated that the model was able to represent the observed OA concentrations in the one site in Beijing. Please resolve this inconsistency or provide more discussion in the text. The discrepancy between the measurements and the simulated concentrations in Figure 4c is large enough that, I do not think the difference in the year could explain it. Also, the symbols were too small to read.

Lines 499-500: "Overall, the model explained most of the observations." I really did not see this in Figure 4c. Please revise and provide an estimate of the bias.

Figure 4: Are there also observational constraints on what fraction of the measured

OC was secondary, e.g. using the EC-tracer method? I think this was in Zhang et al. (2008) and can be shown in Figure 4d for comparison

Lines 509-511: "However, our simulations show in Fig. 5 ... large anthropogenic emissions." How does this statement relate to, or can be used to explain the finding in Fig 4, i.e., the model severely underestimated the observed OC, particularly over China?

Lines 518-519: "In the second ... over China": How does this statement relate to, or can be used to explain the finding in Fig 4, i.e., the model severely underestimated the observed OC, particularly over China?

Figure 7: Again, all of the symbols for the observed values were way too small and unreadable. Please revise.

Figure 9 caption, last line: "over the first domain (top panel) and second domain (bottom panel)": should be 'left panel' and 'right panel', respectively.

Reference Jo et al. (2019), Geosci. Model Dev., 12, 2983-3000.

---

## Author Comment (AC1) · 19 Dec 2020

**We thank the reviewers for the effort to review the manuscript and to provide constructive comments and good suggestions to improve our manuscript. Our replies to the comments and our actions taken to revise the paper (in blue) are given below (the original comments are copied here).**

The modifications corresponding to the comments are labeled in red color and highlighted.

The language and grammar in the revised manuscript have been edited carefully and polished native English speakers by according to the reviewers' comments. (labeled in red color)

Referees' comments:

Referee #1

The microphysical processes of organic aerosol have not yet been well represented, which lead to large uncertainties in current simulation studies. In this article, authors used a new global-regional nested aerosol model combined a particle microphysics module and a volatility basis-set organic aerosol module to simulate microphysical processes of organic aerosol. The model can reproduce the organic aerosol components and the particle number size distribution in Beijing, and spatial distributions of organic carbon and number concentrations of particles condensation larger than 10 nm. They further explored the model's sensitivity to the size distribution of primary emission and volatility distribution of primary organic aerosol. I am glad to see the amount of work presented in the manuscript. This reviewer doesn't find apparent flaw in the method and data the manuscript shows. I think the manuscript can be accepted after the following concerns are addressed.

Reply: Thanks to the reviewer for providing good suggestions to improve our work.

General Comments:

1. The writing is a bit difficult to understanding in many places, which leaves itself open to misinterpretations or confusion, and so the paper could really use a thorough edit from a native English speaker.

Reply: The writing has been edited carefully and polished by native English speakers according to the reviewers' comments. The major revisions are showed in red color in the revised manuscript.

2. To provide a reliable foundation for further analysis, a comprehensive model evaluation including aerosol optical depth, PM$_{2.5}$ is recommended.

Reply: Due to space limitation and continuity of the article, the comparison of annual mean global aerosol optical depth between simulation and MODIS data and PM$_{2.5}$ evaluation in Beijing and surrounding cities are added in the supplement material. The description of PM$_{2.5}$ evaluation in Beijing and surrounding cities is also added in the revised manuscript. (seen in Line 447-449)

Special Comments:

1. Line 85, "also" may be deleted.

2. Line 92-93, "the complication of processes and the mechanisms not well understood" may be replaced with "the unclear complication of processes and the mechanisms".

3. Line 96, "find" should be "found".

4. Line 158,"indicate" should be "indicated".

Reply: The above modifications have been made.

5. Line 266, "primary OA (POA)" should be "POA".

Reply: "POA" is the first appearance and so "primary OA (POA)" is used here.

6. Line 306, "When necessary, SP-LV is redistributed to size-bins ...", please clarify the specific situations.

Reply: The situations to redistribute SP-LV to size-bins include: (1) calculating the particle size in order to simulate the condensation growth and coagulation of secondary particles; (2) the coagulation scavenging of secondary particles by primary particles. The related descriptions were added in the revised manuscript (seen in Line

320-321 ).

7. Please provide the full name for "LV-OA" and "POC" at the first appearance.

Reply: The full name for "LV-OA" were provided in Line 328 and "POC" in Line 330.

8. Line 378, "More details on the observation can be found in the published paper (Du et al., 2017)" may be "More details on the observation can be found in Du et al.(2017)".

Reply: Revised (seen in Line 401-401).

9. Please also provide the correlation coefficients between the observed and simulated BC and simulated POA and observed HOA.

Reply: The correlation coefficients between the observed and simulated BC and simulated POA and observed HOA are showed in Fig.1 (seen in Fig.1) and presented in the revised manuscript (seen in Line 425-426 and 431-432).

10. For figure 4 and 7, the shaded circles are difficult to observe. And the concentrations of secondary organic aerosol and CN10 are recommended to display.

Reply: The observed values of SOA and CN10 are labeled with shaded colors in black circles. The shaded circles have been made clear in Figure 4 and 7. The concentrations of SOA and CN10 are too dense to be clearly displayed, so the exact values are not displayed in Figure 4 and 7.

11. Line 515-516, "The higher concentrations of ASOA than BSOA are also demonstrated by other studies", please some references there.

Reply: The references are added (seen in Line 548).

12. Please give some potential reasons for the differences between spatial

distributions of SV-SOA and LV-SOA.

Reply: The differences between spatial distributions of SV-SOA and LV-SOA are mainly caused by their different formation mechanisms. SV-SOA is mainly from the products of VOCs whereas LV-SOA is from the further oxidation of SV-SOGs. The multi-generation aging processes can make the LV-SOA formed downwind the source regions. Globally, high SV-SOA and LV-SOA concentrations are mainly located in the continental source regions. However, the concentration of LV-SOA is higher than that of SV-SOA in downwind regions. Even over source areas with low emission intensity, such as North America and Europe, LV-SOA also has a higher concentration than does SV-SOA. In the VBS scheme, the organic compounds could undergo the multi-generation aging processes during transport and produce a higher concentration of LV-SOA which mostly remains in particle phase. Consequently, LV-SOA distribution is more homogeneous than SV-SOA does and has a wider spread over the ocean. The reasons for the differences between spatial distributions of SV-SOA and LV-SOA are added in the revised manuscript. (seen in Line 577-582)

13. Line 563, the authors did not provide "Table 3" and "observed values in Fig.6a". Please modify.

Reply: "Table 3" were revised to "Table S1" and "Fig.6a" were revised to "Fig.6a" (seen Line 600 and 601).

14. Figure 7d shows that the high value CN10 is mainly primary over Northeast China where the concentration of secondary organic aerosol is relative high shown in Figure 6. Please explain this phenomenon.

Reply: Though SOA concentration is relative high in North China Plain, they are coated on the primary particles (BC and POC particles) due to the high concentration of primary particles. In our model, primary and secondary particles are distinguished by their physical origin rather than chemical composition (seen in Line 218-221). Even though the concentration of secondary coatings is high, the primary particles with secondary coatings are defined as "primary particles". The large primary emission leads to the high concentration of primary particles (BC and POC; served as the core of "primary particles"), which can scavenge the secondary particles by coagulation and reduce the growth rate of secondary particles by competing for

condensable gases. Therefore, CN10 is dominated by "primary particles" over North China Plain. The corresponding explanation are added in Line 632-634.

15. Line 631, "indicate" should be "indicated".

16. Line 835, "top panel" and "bottom panel" should be "left panel" and "right panel".

Reply: Revised (seen in line 669 and 877).

---

## Author Comment (AC2) · 19 Dec 2020

**We thank the reviewers for the effort to review the manuscript and to provide constructive comments and good suggestions to improve our manuscript. Our replies to the comments and our actions taken to revise the paper (in blue) are given below (the original comments are copied here).**

The modifications corresponding to the comments are labeled in red color and highlighted.

The language and grammar in the revised manuscript have been edited carefully and polished native English speakers by according to the reviewers' comments. (labeled in red color)

Referees' comments:

Referee #2

This manuscript presented a significant effort to couple OA formation pathways and microphysical processes to a global and regional chemical transport model, with the goal of simulating the impacts of OA physics/chemistry to particle number size distribution and mixed particle composition. The work coupled a 1.5-D VBS module and the APM microphysics module to the IAP-AACM chemical transport module. The authors also presented some preliminary comparisons with the observations, and overall the model appeared able to capture the global OA concentrations and the CN10 concentrations. The amount of work done was impressive, and the methods were mostly valid and up-to-date. Overall, I think the paper may be published after clarifying some missing details and improving the figure representation.

Reply: Thanks to the reviewer for the great effort to review our manuscript and to provide constructive suggestions to improve our manuscript.

Section 2.1 Host model: What meteorological data is the IAP-AACM driven by? I see in lines 342 to 350 that the model was driven by meteorological parameters from WRF, but maybe that information can be moved here. Also, how were global meteorological fields obtained from WRF? Did the authors run Global WRF? If so,

then additional references for Global WRF should be included, e.g., Zhang et al. (2012). What was the spatial resolution of the meteorological fields, and was interpolation used? What was the temporal resolution of meteorological fields? I.e., how often were the meteorological fields updated. How was the nudging performed in the three nested domains?

Reply: The IAP-AACM+APM was driven by the global WRF. The essential reference (Zhang et al., 2012) is added in the revised manuscript. The IAP-AACM used the same domain and horizontal grid (i.e., 1° for D01, 0.33° for D02, and 0.11°for D03) as for the global WRF; thus, only vertical interpolation of the meteorological fields of the global WRF was performed to drive the IAP-AACM+APM. The meteorological fields were updated hourly in IAP-AACM+APM. In the first domain, a nudging coefficient of 0.0003 was used in all vertical layers; in the second and third domain, the same nudging coefficient was used in vertical layers except those in boundary layer, where nudging was not used. These necessary description has been added in the revised manuscript (seen in Line 193-194, 360-363, and 367-370).

Zhang, Y., Hemperly, J., Meskhidze, N., and Skamarock, W. C.: The Global Weather Research and Forecasting (GWRF) Model: Model Evaluation, Sensitivity Study, and Future Year Simulation, Atmospheric and Climate Sciences, 02, 231-253, 10.4236/acs.2012.23024, 2012.

Section 2.3 VBS module: I would like to see the model's representation of the relationship between oxidation state and volatility expressed more clearly. Did the authors simply move the oxidation products of POA and IVOC into a volatility bin that is one magnitude lower? What about the fragmented products during the oxidation, i.e., the smaller molecular weight products?

Reply: Considering a single oxidation step would hardly provide enough carbon number reduction required to move the oxidation products of POA and IVOC into a volatility bin that is one magnitude lower. In the 1.5D VBS module (Koo et al., 2014; Yang et al., 2019) used in our study, the POA aging process is approximated by using a "partial conversion" to OOA: Oxidation products of POA are represented as a mixture of POA and OPOA in the next lower volatility bins. For IVOC, lower SOA mass yields are assumed to consider this process. Fragmentation in the VBS module is

implicitly considered through reduction in carbon number of the oxidation products. NOx-dependent product mass yields from oxidation of hydrocarbon precursors were determined based on smog chamber data (Murphy and Pandis, 2009; Hildebrandt et al., 2009). The necessary information has been added in the revised manuscript (seen in Line 278-281, 285-287, and 296-299).

*Murphy, B., and Pandis, S.: Simulating the Formation of Semivolatile Primary and Secondary Organic Aerosol in a Regional Chemical Transport Model, Environmental science & technology, 43, 4722-4728, 2009.*

*Hildebrandt, L., Donahue, N.M., Pandis, S.N.: High formation of secondary organic aerosol from the photo-oxidation of toluene. Atmos. Chem. Phys. 9, 2973-2986, 2009.*

*Koo, B., Knipping, E., and Yarwood, G.: 1.5-Dimensional volatility basis set approach for modeling organic aerosol in CAMx and CMAQ, Atmospheric Environment, 95, 158–164, 10.1016/j.atmosenv.2014.06.031, 2014.*

*Yang, W., Li, J., Wang, W., Li, J., Ge, M.-F., Sun, Y., Chen, G., ge, B., Tong, S., Wang, Q., and Wang, Z.: Investigating secondary organic aerosol formation pathways in China during 2014, Atmospheric Environment, 213, 10.1016/j.atmosenv.2019.05.057, 2019.*

Section 2.3 VBS module: Also, a recent paper (Jo et al., 2019) indicated that the VBS representation of SOA formation from isoprene is incorrect because the reactive uptake pathway dominates SOA formation from isoprene. Please discuss this point, the lack of reactive update pathways in this model, and the implication for the present model results.

Reply: Yes, based on the comparison of full-chemistry calculation and VBS simulation, the VBS representation could not capture the physicochemical dependencies of SOA formation on dominant pathway from isoprene and VBS may underestimate the biogenic SOA from isoprene (Jo et al., 2019). Nevertheless, the biogenic SOA could not explain the SOA concentration (Spracklen et al. 2011; Matsui et al., 2014; Lin et al., 2016). Therefore, the important roles of anthropogenic SOA revealed in our study still hold true and the major conclusions would not change. However, the fixed parameters in VBS make it difficult to represent the real formation pathway of SOA and capture the response of SOA to emission changes. More accurate parameters considering the key physicochemical dependencies should be incorporated to update the VBS module in our model. This is critical to (1) accurately quantify the contribution of biogenic and anthropogenic sources to OA, (2) evaluate the effectiveness of control measures to reduce OA concentration in order to improve

air quality, and (3) explore the aerosol-climate-vegetation interactions. Thank the reviewer for this valuable comment and notice for our future work. The necessary discussions have been added in our revised manuscript (seen in Line 543-547 and 821-825).

*Jo, D. S., Hodzic, A., Emmons, L. K., Marais, E. A., Peng, Z., Nault, B. A., Hu, W. W., Campuzano-Jost, P., and Jimenez, J. L.: A simplified parameterization of isoprene-epoxydiol-derived secondary organic aerosol (IEPOX-SOA) for global chemistry and climate models: a case study with GEOS-Chem v11-02-rc, Geoscientific Model Development, 12, 2983-3000, 10.5194/gmd-12-2983-2019, 2019.*

Lines 359-360: "In the LV_POA and HV_POA experiment, quartiles of the abovementioned distribution factors are used". Not sure what this meant. Looking at Table 2, I do not see the use of 'quartiles of the above-mentioned factors'. What different factors were used in the LV_POA and HV_POA experiments, respectively?

Reply: The different factors used in the LV_POA and HV_POA experiments were provided in the "Volatility distribution" column in Table 2. The quartiles of POA volatility factor are taken from May et al. (2003a, b, c). Modification were made in Line 383-384.

---

## Author Comment (AC4) · 5 Jan 2021

**We thank the reviewers for the effort to review the manuscript and to provide constructive comments and good suggestions to improve our manuscript. Our replies to the comments and our actions taken to revise the paper (in blue) are given below (the original comments are copied here).**

The modifications corresponding to the comments are labeled in red color and highlighted.

The language and grammar in the revised manuscript have been edited carefully and polished native English speakers by according to the reviewers' comments. (labeled in red color)

Referees' comments:

Referee #2

This manuscript presented a significant effort to couple OA formation pathways and microphysical processes to a global and regional chemical transport model, with the goal of simulating the impacts of OA physics/chemistry to particle number size distribution and mixed particle composition. The work coupled a 1.5-D VBS module and the APM microphysics module to the IAP-AACM chemical transport module. The authors also presented some preliminary comparisons with the observations, and overall the model appeared able to capture the global OA concentrations and the CN10 concentrations. The amount of work done was impressive, and the methods were mostly valid and up-to-date. Overall, I think the paper may be published after clarifying some missing details and improving the figure representation.

Reply: Thanks to the reviewer for the great effort to review our manuscript and to provide constructive suggestions to improve our manuscript.

Section 2.1 Host model: What meteorological data is the IAP-AACM driven by? I see in lines 342 to 350 that the model was driven by meteorological parameters from WRF, but maybe that information can be moved here. Also, how were global meteorological fields obtained from WRF? Did the authors run Global WRF? If so,

then additional references for Global WRF should be included, e.g., Zhang et al. (2012). What was the spatial resolution of the meteorological fields, and was interpolation used? What was the temporal resolution of meteorological fields? I.e., how often were the meteorological fields updated. How was the nudging performed in the three nested domains?

Reply: The IAP-AACM+APM was driven by the global WRF. The essential reference (Zhang et al., 2012) is added in the revised manuscript. The IAP-AACM used the same domain and horizontal grid (i.e., 1° for D01, 0.33° for D02, and 0.11°for D03) as for the global WRF; thus, only vertical interpolation of the meteorological fields of the global WRF was performed to drive the IAP-AACM+APM. The meteorological fields were updated hourly in IAP-AACM+APM. In the first domain, a nudging coefficient of 0.0003 for wind, temperature, and water vapor was used in all vertical layers; in the second and third domain, the same nudging scheme was used in vertical layers except those in boundary layer, where nudging was not used. These necessary description has been added in the revised manuscript (seen in Line 192-194, 362-365, and 369-372).

*Zhang, Y., Hemperly, J., Meskhidze, N., and Skamarock, W. C.: The Global Weather Research and Forecasting (GWRF) Model: Model Evaluation, Sensitivity Study, and Future Year Simulation, Atmospheric and Climate Sciences, 02, 231-253, 10.4236/acs.2012.23024, 2012.*

Section 2.3 VBS module: I would like to see the model's representation of the relationship between oxidation state and volatility expressed more clearly. Did the authors simply move the oxidation products of POA and IVOC into a volatility bin that is one magnitude lower? What about the fragmented products during the oxidation, i.e., the smaller molecular weight products?

Reply: Considering a single oxidation step would hardly provide enough carbon number reduction required to move the oxidation products of POA and IVOC into a volatility bin that is one magnitude lower. In the 1.5D VBS module (Koo et al., 2014; Yang et al., 2019) used in our study, the POA aging process is approximated by using a "partial conversion" to OOA: Oxidation products of POA are represented as a mixture of POA and OPOA in the next lower volatility bins. For IVOC, lower SOA mass yields are assumed to consider this process. Fragmentation in the VBS module is

implicitly considered through reduction in carbon number of the oxidation products. NOx-dependent product mass yields from oxidation of hydrocarbon precursors were determined based on smog chamber data (Murphy and Pandis, 2009; Hildebrandt et al., 2009). The necessary information has been added in the revised manuscript (seen in Line 278-281, 285-287, and 298-301).

*Murphy, B., and Pandis, S.: Simulating the Formation of Semivolatile Primary and Secondary Organic Aerosol in a Regional Chemical Transport Model, Environmental science & technology, 43, 4722-4728, 2009.*

*Hildebrandt, L., Donahue, N.M., Pandis, S.N.: High formation of secondary organic aerosol from the photo-oxidation of toluene. Atmos. Chem. Phys. 9, 2973-2986, 2009.*

*Koo, B., Knipping, E., and Yarwood, G.: 1.5-Dimensional volatility basis set approach for modeling organic aerosol in CAMx and CMAQ, Atmospheric Environment, 95, 158–164, 10.1016/j.atmosenv.2014.06.031, 2014.*

*Yang, W., Li, J., Wang, W., Li, J., Ge, M.-F., Sun, Y., Chen, G., ge, B., Tong, S., Wang, Q., and Wang, Z.: Investigating secondary organic aerosol formation pathways in China during 2014, Atmospheric Environment, 213, 10.1016/j.atmosenv.2019.05.057, 2019.*

Section 2.3 VBS module: Also, a recent paper (Jo et al., 2019) indicated that the VBS representation of SOA formation from isoprene is incorrect because the reactive uptake pathway dominates SOA formation from isoprene. Please discuss this point, the lack of reactive update pathways in this model, and the implication for the present model results.

Reply: Yes, based on the comparison of full-chemistry calculation and VBS simulation, the VBS representation could not capture the physicochemical dependencies of SOA formation on dominant pathway from isoprene and VBS may underestimate the biogenic SOA from isoprene (Jo et al., 2019). Nevertheless, the biogenic SOA could not explain the SOA concentration over China (Spracklen et al. 2011; Matsui et al., 2014; Lin et al., 2016). Therefore, the important roles of anthropogenic SOA revealed in our study still hold true and the major conclusions would not change. However, the fixed parameters in VBS make it difficult to represent the real formation pathway of SOA and capture the response of SOA to emission changes. More accurate parameterizations considering the key physicochemical dependencies should be incorporated to update the VBS module in our model. This is critical to (1) accurately quantify the contribution of biogenic and anthropogenic sources to OA, (2) evaluate the effectiveness of control measures to

reduce OA concentration in order to improve air quality, and (3) explore the aerosol-climate-vegetation interactions. Thank the reviewer for this valuable comment and notice for our future work. The necessary discussions have been added in our revised manuscript (seen in Line 568-571 and 852-856).

*Jo, D. S., Hodzic, A., Emmons, L. K., Marais, E. A., Peng, Z., Nault, B. A., Hu, W. W., Campuzano-Jost, P., and Jimenez, J. L.: A simplified parameterization of isoprene-epoxydiol-derived secondary organic aerosol (IEPOX-SOA) for global chemistry and climate models: a case study with GEOS-Chem v11-02-rc, Geoscientific Model Development, 12, 2983-3000, 10.5194/gmd-12-2983-2019, 2019.*

Lines 359-360: "In the LV_POA and HV_POA experiment, quartiles of the abovementioned distribution factors are used". Not sure what this meant. Looking at Table 2, I do not see the use of 'quartiles of the above-mentioned factors'. What different factors were used in the LV_POA and HV_POA experiments, respectively?

Reply: The different factors used in the LV_POA and HV_POA experiments were provided in the "Volatility distribution" column in Table 2. The quartiles of POA volatility factor are taken from May et al. (2003a, b, c) and Robinson et al. (2007). Modification are made in Line 390-393.

Table 2 and related text on the design of the sensitivity experiment: overall, I think the sensitivity experiment could be explained more clearly. I was not able to understand what was the goals of the sensitivity experiments and how those goals relate to the parameters in Table 2.

Reply: Because the size distribution of primarily emitted particles and the volatility distribution of POA have substantial impacts on the simulation of particle number concentration over areas influenced by anthropogenic sources, the sensitivity experiments were designed to investigate the impacts of these factors on our study results. The description of sensitivity experiments and their relations with the parameters in Table 2 are described more clearly in the revised manuscript (seen in Sect.3.2).

Lines 382- 383: "...the China Atmosphere Watch Network... Zhang et al. (2008)": What year(s) were the measurements? The writing of this sentence seemed to suggest that the measurements were from 2010, which cannot be possible.

Reply: Yes, the year of OC measurements in China was 2006. "2010" is changed to "2006" in the revised manuscript (seen in Line 414).

Lines 437 to 439: "The number concentration of particles from 100 nm to 1000 nm ...correlation coefficient being 0.70.': Figure 2 uses a different unit for particle size (micrometer), and I do not see the diameter extending to 1000 nm. Also, how were the normalized bias and the correlation coefficient calculated? Did the authors calculated only the bias and correlation for the time series of the total number concentration (which is not shown)? Or did they calculated a mean bias and correlation for the entire PNSD spectrum? If the latter, how was this done, and did the statistics entail a preferential weighting of the smaller particles?

Reply: Yes, the shaded figure (Fig.2) did not extending to 1000 nm. Because the size bins of observation and the model are different, the particle number size distribution (PNSD) of observation is mapped to the size bins of the model when comparing the particle number size distribution in Fig.2. The number concentration of particles from 100 nm to 1000 are calculated by adding the number concentration in the size bin from 100 nm to 1000 nm. Due to the limitation of detection size range (15-685nm) of the measurement, the diameter shown in Fig.2 was not extended to 1000 nm. In the community, "100 nm to 1000 nm" is commonly used as a proxy of accumulation mode and the particles from 100 nm to 685 nm can account for the most of particles number concentration from 100 nm to 1000 nm. Due to these reasons, "100 nm to 1000 nm" was used in our study. However, this expression is inaccurate. In the revised manuscript, "100 nm to 1000 nm" are changed to the size range of measurements and the comparison method of simulated particle number concentration with observations is expressed clearly (seen in Line 412-414 and 477).

Figure 4: The measurements were too small and unreadable in this figure. Please enlarge and circle with a black or white outline.

Reply: The circles are made clear with a black outline in the revised manuscript (seen in Fig.4).

Figure 4: Also, the OC measurements in China all appeared to be much, much higher

than the simulated concentrations. This is inconsistent with what was shown in Figure 1, where the authors indicated that the model was able to represent the observed OA concentrations in the one site in Beijing. Please resolve this inconsistency or provide more discussion in the text. The discrepancy between the measurements and the simulated concentrations in Figure 4c is large enough that, I do not think the difference in the year could explain it. Also, the symbols were too small to read.

Reply: The emission data for 2010 simulation (Fig.4) was different from the emission used in the simulation during the period from August 22 to September 30, 2015. 2015 case used the multi-resolution emission inventory for China in 2015 (http://www.meicmodel.org). 2010 simulation used a publicly available datasets for 2010 (https://edgar.jrc.ec.europa.eu/htap_v2/index.php). The difference of BC and OC emission between 2010 and 2015 over China is displayed in Fig.R1. Compared with the emission in 2010, a reduced emission of BC and OC in main source regions can be seen in 2015 over China. In our study, the total OC emission in 2010 over China was 3.54 Tg C yr$^{-1}$. Using the GEOS-Chem and a emission of 3.95 Tg C yr$^{-1}$ for OC, Fu et al. (2012) found the model underestimated OC at most observation sites, particularly in January in 2006. Their top-down estimation for total OC emission was 6.67 Tg C yr$^{-1}$, greatly higher than the OC emission in our study. Studies indicated that the uncertainties of BC and OC emission can be higher than 200% (Zhao et al., 2013; Li et al., 2015). We agree that the underestimation of OC in 2010 could not be attributed only to the uncertainties in emission. However, the difference of emission is a major factor responsible for the different model performance in simulating OC concentration between 2010 and 2015. According to the suggestions of the reviewer, some discussions on the inconsistent model performance between 2010 and 2015 are added (seen in Line 545-549). In addition, the explanations for OC underestimation are presented (seen in Line 536-545). The symbols in Fig.4 are made clear to read.

[Figure]

Fig.R1 Differences of (a) BC and (b) OC emission rates between 2010 and 2015
(2015-2010)

*Fu, T. M., Cao, J. J., Zhang, X. Y., Lee, S. C., Zhang, Q., Han, Y. M., Qu, W. J., Han, Z., Zhang, R., Wang, Y. X., Chen, D., and Henze, D. K.: Carbonaceous aerosols in China: top-down constraints on primary sources and estimation of secondary contribution, Atmospheric Chemistry and Physics, 12, 2725-2746, 10.5194/acp-12-2725-2012, 2012.*

*Li, M., Liu, H., Geng, G. N., Hong, C. P., Liu, F., Song, Y., Tong, D., Zheng, B., Cui, H. Y., Man, H. Y., Zhang, Q., and He, K. B.: Anthropogenic emission inventories in China: a review, Natl Sci Rev, 4, 834-866, 10.1093/nsr/nwx150, 2017.*

*Zhao, Y., Zhang, J., and Nielsen, C. P.: The effects of recent control policies on trends in emissions of anthropogenic atmospheric pollutants and CO2 in China, Atmospheric Chemistry and Physics, 13, 487-508, 10.5194/acp-13-487-2013, 2013.*

*Zhang, X. Y., Wang, J. Z., Wang, Y. Q., Liu, H. L., Sun, J. Y., and Zhang, Y. M.: Changes in chemical components of aerosol particles in different haze regions in China from 2006 to 2013 and contribution of meteorological factors, Atmospheric Chemistry and Physics, 15, 12935-12952, 10.5194/acp-15-12935-2015, 2015.*

Lines 499-500: "Overall, the model explained most of the observations." I really did not see this in Figure 4c. Please revise and provide an estimate of the bias.

Reply: Yes, the model underestimated the OC concentration over China. Nevertheless, the spatial variations of OC at different observation sites, particularly the west-east gradient, were well reproduced. The OC concentration in the United States were also reasonably simulated (Fig.4a). According to the comment, the inappropriate expressions are modified (seen in Line 531-534). The underestimation of OC at sites in China can be explained by the following reasons: (1) the underestimation of OC emission; (2) representativeness difference between observation and simulation results; (3) uncertainties in model mechanism calculating OC concentration.

The study of Zhao et al. (2012) and Li et al. (2017) pointed out that the emission of OC over China did not change much. The difference of OC concentration between 2006 and 2010 is also small (Zhang et al., 2015). The OC emission (3.54 Tg C $yr^{-1}$) in our study is 47% lower than the estimated emission (6.67 Tg C $yr^{-1}$) in Fu et al.

(2012). In addition, in the VBS module used in our study, POA emissions were distributed to the five volatility bins, which can lead to an underestimation of OC and OA from POA emission (Donahue et al., 2009). Following the recommendation of existing study (Tsimpidi et al., 2010; Shrivastava et al., 2011), we did the experiment increasing the existing POA emission by a factor of 3. Although the underestimation of OC in China was reduced, the OC concentrations in the United States were overestimated (Fig.R3). Considering the substantial contribution of intermediate volatility organic compounds (IVOC) to SOA and OA (Zhao et al., 2016; Yang et al., 2019), and the underestimation of SOC fraction in China, the IVOC emissions used in our study may be underestimated. Based on these facts, the underestimation of OC emission over China should largely be responsible for the underestimated OC in China.

However, the underestimation of SOC fraction also indicates to the deficiency of SOA formation mechanism. Although studies (Zhao et al., 2016; Yang et al., 2019) have suggested that IVOCs have a large contribution to SOA and OA over China, the emissions of IVOCs were not included in the traditional emission inventory, which make it difficult to estimate SOA from IVOCs. The underestimation of their emission and production yields to SOA can lead to considerable underestimation of SOA and OA. Aqueous-phase formation processes of SOA have an evident influence on the particle properties and total SOA mass (Ervens et al., 2011). Nevertheless, aqueous-phase processes of SOA was not included and it may cause an underestimation of SOA and OA.

According to the reviewer's comment, the reasonable description, necessary discussion, and the model bias were added in the revised manuscript (seen in Line 532-534, 545-549, and 586-590).

[Figure]

Fig.R2 Surface layer horizontal spatial distributions of organic carbon concentrations. The shaded circles denote the observed concentrations.

*Donahue, N., Robinson, A., and Pandis, S.: Atmospheric organic particulate matter: From smoke to secondary organic aerosol, Atmospheric Environment, 43,*

*94-106, 10.1016/j.atmosenv.2008.09.055, 2009.*

*Ervens, B., Turpin, B. J., and Weber, R. J.: Secondary organic aerosol formation in cloud droplets and aqueous particles (aqSOA): a review of laboratory, field and model studies, Atmos. Chem. Phys., 11, 11069–11102, https://doi.org/10.5194/acp-11-11069-2011, 2011.*

*Shrivastava, M., Fast, J., Easter, R., Gustafson, W. I., Zaveri, R. A., Jimenez, J. L., Saide, P., and Hodzic, A.: Modeling organic aerosols in a megacity: comparison of simple and complex representations of the volatility basis set approach, Atmospheric Chemistry and Physics, 11, 6639-6662, 10.5194/acp-11-6639-2011, 2011.*

*Tsimpidi, A. P., Karydis, V. A., Zavala, M., Lei, W., Molina, L., Ulbrich, I. M., Jimenez, J. L., and Pandis, S. N.: Evaluation of the volatility basis-set approach for the simulation of organic aerosol formation in the Mexico City metropolitan area, Atmospheric Chemistry and Physics, 10, 525-546, DOI 10.5194/acp-10-525-2010, 2010.*

*Yang, W., Li, J., Wang, W., Li, J., Ge, M.-F., Sun, Y., Chen, G., ge, B., Tong, S., Wang, Q., and Wang, Z.: Investigating secondary organic aerosol formation pathways in China during 2014, Atmospheric Environment, 213, 10.1016/j.atmosenv.2019.05.057, 2019.*

*Zhao, B., Wang, S., Donahue, N., Jathar, S., Huang, X., Wu, W., Hao, J., and Robinson, A.: Quantifying the effect of organic aerosol aging and intermediate-volatility emissions on regional-scale aerosol pollution in China, Scientific reports, 6, 10.1038/srep28815, 2016a.*

Figure 4: Are there also observational constraints on what fraction of the measured OC was secondary, e.g. using the EC-tracer method? I think this was in Zhang et al. (2008) and can be shown in Figure 4d for comparison.

Reply: SOC fractions in Zhang et al. (2008) are shown in Fig.4d.

Lines 509-511: "However, our simulations show in Fig. 5 ... large anthropogenic emissions." How does this statement relate to, or can be used to explain the finding in Fig 4, i.e., the model severely underestimated the observed OC, particularly over

China? Lines 518-519: "In the second ... over China": How does this statement relate to, or can be used to explain the finding in Fig 4, i.e., the model severely underestimated the observed OC, particularly over China?

Reply: In the case of lower simulated OC concentration than observation, the underestimation of SOC in major anthropogenic source regions indicated that the model underestimated the formation of anthropogenic SOA (ASOA) in these regions. Underestimation of AOA should be responsible for the underestimation of OA over China. The exact contribution of ASOA to SOA would be greater than estimated in our simulation. The importance of ASOA is still true though our model underestimated the absolute concentration of OC, particularly over China. However, the exact contribution of ASOA should be refined by incorporating the precise emission including non-traditional emission and improving the representation of SOA formation in the VBS module in our future study. The relations between the underestimation of OC and the higher concentration of AOA than BOA are added and the influences of the OC underestimation, particularly over China, are added in our revised manuscript (seen in Line 586-590).

Figure 7: Again, all of the symbols for the observed values were way too small and unreadable. Please revise.

Reply: revised (seen in Fig.7).

Figure 9 caption, last line: "over the first domain (top panel) and second domain (bottom panel)": should be 'left panel' and 'right panel', respectively.

Reply: revised (seen in line 909).

Reference Jo et al. (2019), Geosci. Model Dev., 12, 2983-3000.

---

## Author Response (AR2)

We thank the editors and reviewers for the effort to review the manuscript and to provide good comments and suggestions to improve our manuscript. Our replies to the comments and our actions taken to revise the paper (in blue) are given below (the original comments are copied here).

The modifications are marked in red color and highlighted.

Comments:

The authors have comprehensively and convincingly replied to all reviewer comments and provided a much improved revised manuscript. There are only 2 minor revisions required. These relate to clarifications of the two revisions inserted at lines 278-281 and 285-287 respectively. I cannot quite understand either sentence.

Does the first sentence mean: "A single oxidation step would be unable to sufficiently reduce carbon number for the oxidation products of POA and IVOC to be one order of magnitude higher in volatility"? (noting that a reduction in carbon number increases the volatility)

and the sentence second mean: "To account for the insufficient reduction in carbon number of the IVOC product, reduction in SOA mass yields from IVOC are assumed..."?

Reply: Yes, a reduction in carbon number increases the volatility when other factors are not taken into account. The sentence in our manuscript caused misunderstanding. In the 2-D VBS space, the reaction trajectory of POA aging initially follows the carbon-number isopleths (oxygenation) but then transitions to fragmentation of more highly-oxygenated products (Donahue et al., 2012). In the 1.5-D VBS, reduction in carbon number indicates that fragmentation is implicitly accounted for. Considering that a single oxidation step would be unable to sufficiently transfer OA from HOA basis to OOA basis, the concept of "partial conversion" is used in 1.5-D VBS; that is, the oxidation products are a mixture of POA and oxidized POA (OPOA) in the adjacent lower volatility bins (Koo et al., 2014). In the oxidation process, although the averaged carbon number of the organic mixture can be reduced by fragmentation, the oxygen is added and the volatility of major oxidation products is decreased.

To make the meaning more clear, the sentence in line 278-281 is changed to

"Considering a single oxidation step would not be able to move the oxidation products of POA into the oxidized OA basis in the volatility bin that is one magnitude lower" (seen in line 278-280).

The sentence "To consider insufficient carbon number reduction of IVOC product, a lower SOA mass yields from IVOC are assumed" is changed to "To account for the insufficient reduction in carbon number and volatility decrease of the IVOC product, the SOA mass yields from IVOC are assumed to be lower than that of POA" (seen in line 285-287).